# DIVERSE: Disagreement-Inducing Vector Evolution for Rashomon Set Exploration

**Gilles Eerlings, Brent Zoomers, Jori Liesenborgs, Gustavo Rovelo Ruiz & Kris Luyten**
Digital Future Lab - Flanders Make
UHasselt - Hasselt University
Diepenbeek, Belgium
`{firstname.lastname}@uhasselt.be`

## Abstract

We propose DIVERSE, a framework for systematically exploring the Rashomon set of deep neural networks, the collection of models that match a reference model's accuracy while differing in their predictive behavior. DIVERSE augments a pretrained model with Feature-wise Linear Modulation (FiLM) layers and uses Covariance Matrix Adaptation Evolution Strategy (CMA-ES) to search a latent modulation space, generating diverse model variants without retraining or gradient access. Across MNIST, PneumoniaMNIST, and CIFAR-10, DIVERSE uncovers multiple high-performing yet functionally distinct models. Our experiments show that DIVERSE offers a competitive and efficient exploration of the Rashomon set, making it feasible to construct diverse sets that maintain robustness and performance while supporting well-balanced model multiplicity. While retraining remains the baseline for generating Rashomon sets, DIVERSE achieves comparable diversity at reduced computational cost.

## 1 Introduction

In many machine learning tasks, there exists a large set of diverse models that achieve a similar performance for the same tasks, yet differ substantially in the pathway leading to their predictions. These differences can manifest in decision boundaries, internal representations, or feature importance (Fisher et al., 2019; Black et al., 2022). This phenomenon, where models attain similar performance yet differ in their decision pathways, is known as the Rashomon Effect (Breiman, 2001), often also referred to as model multiplicity (Black et al., 2022; Dai et al., 2025). The collection of these functionally diverse yet accurate models is known as a Rashomon set (Semenova et al., 2022).

Within a Rashomon set, models may produce divergent predictions for identical inputs, a property termed predictive multiplicity (Marx et al., 2020). These disagreements are not errors, but valid variations within the model family, revealing that equally performant models can still differ on specific samples. This variation highlights when predictions depend on the model rather than the data. This has been leveraged for tasks such as assessing model uncertainty (Eerlings et al., 2025), promoting fairness (Dai et al., 2025), improving interpretability (Semenova et al., 2022), and increasing model flexibility (Jain et al., 2025). At the same time, multiplicity poses practical challenges in high-stakes applications, since equally accurate models may assign different outcomes which might be considered a threat to fairness and trust in machine learning (Meyer et al., 2025). Rashomon sets capture the hidden flexibility of models while highlighting the need for systematic, computationally feasible methods to explore them.

While the Rashomon Effect is better understood in simpler model families such as decision trees (Xin et al., 2022) and generalized additive models (Zhong et al., 2023), its characterization in deep neural networks remains limited. The challenge is to explore their vast hypothesis space, while staying within a narrow margin of optimal performance, making Rashomon set exploration both computationally demanding and conceptually complex. Recent work has approximated Rashomon sets in neural networks through model retraining (Ganesh, 2024; Eerlings et al., 2025), Adversarial Weight Perturbation (AWP) (Hsu & Calmon, 2022), and dropout-based sampling (Hsu et al., 2024). These approaches offer valuable insights and a strong foundation for our work, in which we ex-

plore important trade-offs: retraining is computationally costly, AWP can scale poorly, and dropout sampling offers limited control over diversity. Building on previous work, our work proposes an alternative perspective that emphasizes efficiency and explicit diversity control.

We propose DIVERSE, a gradient-free framework for systematically exploring a local Rashomon set of deep neural networks without costly retraining. By augmenting the pretrained model with Feature-wise Linear Modulation (FiLM) layers (Perez et al., 2018), we perform a controlled search of the local environment around the model by modulating the pre-activations using a latent vector $z$. This design enables fine-grained adjustments to internal representations while leaving the original weights untouched. By varying $z$, the model traverses a modulation space, a low-dimensional region of the hypothesis space where functionally distinct variants can be realized without retraining. To search this space, we employ Covariance Matrix Adaptation Evolution Strategy (CMA-ES) (Hansen, 2006), which optimizes over candidate vectors $z$ to induce predictive disagreement with the pre-trained model while maintaining accuracy within a predefined margin, thereby delineating the Rashomon set.

**Contributions.** (1) We introduce a robust Rashomon exploration method that combines FiLM-based modulation with CMA-ES, requiring no retraining or gradient access; (2) on MNIST, PneumoniaMNIST, and CIFAR-10, DIVERSE discovers accurate and diverse model sets, outperforming retraining in efficiency and dropout most of the time in diversity.

## 2 BACKGROUND & RELATED WORK

**Defining and Approximating Rashomon Sets.** We consider the standard supervised learning setting with dataset $\mathcal{D} = \{(x_i, y_i)\}_{i=1}^{N}$, inputs $x_i \in \mathcal{X} \subseteq \mathbb{R}^q$ and corresponding labels $y_i \in \mathcal{Y} \subseteq \mathbb{R}^k$, and a hypothesis space $\mathcal{H}$ where each model $f_w \in \mathcal{H}$ is parameterized by weights $w \in \mathbb{R}^p$. Performance of each model is evaluated by a loss function $L$, with empirical risk, following the principle of Empirical Risk Minimization (Vapnik, 1995):

$$\hat{\mathcal{R}}(f_w) = \frac{1}{N} \sum_{i=1}^{N} L(f_w(\boldsymbol{x}_i), \boldsymbol{y}_i). \tag{1}$$

Standard supervised learning seeks a model that minimizes empirical risk, obtained through standard training, which we denote as the reference model $f_{ref}$. However, multiple parameter configurations $w$ can minimize the empirical risk equally well, yielding hypotheses $f_w \in \mathcal{H}$ that match the performance of $f_{ref}$. The existence of many such solutions gives rise to the Rashomon set $\mathcal{R}_\epsilon$, defined as the subset of models whose empirical risk lies within an $\epsilon$-tolerance, known as the Rashomon parameter (Semenova et al., 2022; 2023), of the reference model, for $\epsilon \geq 0$ (Fisher et al., 2019).

Enumerating the complete Rashomon set is computationally infeasible in the high-dimensional hypothesis space of deep neural networks. Following Hsu et al. (2024), we therefore work with an *empirical Rashomon set*, defined over the finite collection of $m$ models produced during search:

$$\mathcal{R}_\epsilon^m = \{f_{w_i} \in \mathcal{H}_m \mid \hat{\mathcal{R}}(f_{w_i}) \leq \hat{\mathcal{R}}(f_{ref}) + \epsilon\}, \tag{2}$$

where $\mathcal{H}_m \subset \mathcal{H}$ is the finite candidate pool. As $m$ grows, $\mathcal{R}_\epsilon^m$ increasingly approximates $\mathcal{R}_\epsilon$ (Hsu et al., 2024). For brevity, we refer to $\mathcal{R}_\epsilon^m$ simply as the Rashomon set in the remainder of this paper. The Rashomon Ratio (Semenova et al., 2022) summarizes this set by measuring the fraction of candidates in $\mathcal{H}_m$ that meet the constraint, providing a scalar view of Rashomon set size.

Several strategies have been proposed to approximate Rashomon sets in deep neural networks. Retraining from scratch with different seeds, hyperparameters, or augmentations (Ganesh, 2024; Eerlings et al., 2025) offers global exploration but is computationally costly and does not always guarantee models with equivalent performance. Adversarial Weight Perturbation (Hsu & Calmon, 2022) perturbs a trained model's weights to induce disagreement while preserving accuracy, but this requires costly optimization for each (sample, class) pair, making it way slower than retraining and impractical for large neural networks (Hsu et al., 2024). Dropout-based sampling (Hsu et al., 2024) provides a scalable, training-free alternative by stochastically generating subnetworks at test time, while showing substantial speedups over retraining, the resulting diversity is limited and not explicitly controlled.

**FiLM Layers for Latent Modulation.** Feature-wise Linear Modulation (FiLM) (Perez et al., 2018) is a conditioning mechanism that applies affine transformations to neural activations, enabling controlled modulation of intermediate representations. A FiLM layer has the form

$$FiLM(h; \gamma; \beta) = \gamma \odot h + \beta, \tag{3}$$

where $h$ is a hidden pre-activation, and $\gamma$, $\beta$ are the modulation parameters that may be dynamically computed by another function.

FiLM has been widely adopted for conditional computation across a range of settings. In domain generalization, MixStyle (Zhou et al., 2021) introduces stochastic affine perturbations, which are FiLM-like, to simulate domain shifts. In few-shot learning, Tseng et al. (2020) use feature-wise transformations to adapt representations to new tasks. A related use is found in FiLM-Ensemble (Turkoglu et al., 2022), where ensemble members are defined by distinct, learned FiLM parameters optimized for epistemic uncertainty.

In our work, we repurpose FiLM layers to define a family of modulated models centered around a fixed reference network ($f_{\text{ref}}$). Rather than learning new weights for each instance, we freeze the network and FiLM projection layers, and search over a latent vector $z$ from which $\gamma$ and $\beta$ are derived. This defines a continuous subspace of the hypothesis space, the modulation space, enabling controlled exploration without retraining. We denote the model induced by latent vector $z$ as $f_z$.

**CMA-ES for Diversity-Oriented Search.** Covariance Matrix Adaptation Evolution Strategy (CMA-ES) is a widely used optimization algorithm for continuous, non-convex search spaces (Hansen, 2006). Unlike gradient-based methods, it does not require access to objective gradients. Instead, it adapts a multivariate Gaussian distribution over the search space by repeatedly sampling candidate solutions, evaluating their fitness, and updating the distribution parameters based on the relative ranking of candidates. Specifically, candidates at generation $g$ are drawn as:

$$x_k^{(g)} \sim \mathcal{N}(m^{(g)}, \sigma^{(g)^2} C^{(g)}), \tag{4}$$

where $m^{(g)}$ is the current best guess (or mean of the search space), $\sigma^{(g)}$ is the global step size, and $C^{(g)}$ is the covariance matrix that shapes the search distribution. These parameters are updated across generations based on the relative fitness of sampled candidates. The mean is shifted toward better-performing solutions, the covariance matrix is adapted to reflect the directions of successful search steps, and the step size is adjusted using the length of a cumulative path.

Because CMA-ES adapts a full covariance matrix, it can capture correlations between variables, a crucial property in our setting, where the FiLM-modulated latent vector $z$ induces a non-separable landscape: each coordinate of $z$ affects many FiLM layers simultaneously, leading to strong interactions across dimensions. Full-covariance CMA-ES is designed for such coupled structures through its rotational invariance and ability to learn arbitrary covariance patterns during the search (Hansen, 2006). Empirical comparisons confirm this: CMA-ES outperforms alternative evolutionary strategies, such as Differential Evolution and Particle Swarm Optimization, on low- to moderate-dimensional non-separable or ill-conditioned problems (Auger et al., 2009; Omidvar & Li, 2011), making it a suitable optimizer for exploring our modulation space

However, full-covariance CMA-ES becomes expensive in high dimensions (e.g., over several hundred dimensions) (Omidvar & Li, 2011). In our work, we leverage these strengths, namely its ability to capture correlations, handle non-separable landscapes, and learn rich covariance structures, in the context of latent vector $z$ search, while considering the scalability limits of CMA-ES, which will influence our choice of latent dimensionality.

**Quantifying Diversity.** To explore distinct solutions in the $z$-space, we require measures that quantify functional differences between models. We distinguish two complementary notions:

*Hard disagreement.* A standard way to quantify diversity between two classifiers is to measure the proportion of inputs on which their predicted labels differ. Formally, for two models $f_{w_a}$ and $f_{w_b}$ evaluated on dataset $\mathcal{D}$, hard disagreement is given by:

$$\text{Dis}(f_{w_a}, f_{w_b}) = \frac{1}{N} \sum_{i=1}^{N} \mathbb{1}[\hat{y}_i^a \neq \hat{y}_i^b], \quad \hat{y}_i^a = f_{w_a}(x_i), \hat{y}_i^b = f_{w_b}(x_i), \tag{5}$$

where $\mathbb{1}[\cdot]$ denotes the indicator function, equal to 1 if the condition inside holds and 0 otherwise. This notion of disagreement originates in ensemble learning, where it was introduced as a measure of classifier diversity (Skalak, 1996; Ho, 1998; Kuncheva & Whitaker, 2003).

*Soft disagreement.* To quantify differences between two models' predictive behaviors, we compare their output probability distributions on the same inputs. While Kullback-Leibler (KL) (Kullback & Leibler, 1951) and Jensen-Shannon (JS) (Lin, 1991) divergences are common choices for measuring distributional differences, both rely on logarithmic ratios that assume overlapping support between distributions. This assumption becomes problematic when models produce near-deterministic outputs, leading to zero probabilities and numerical instabilities (Thomas M. Cover, 2005). We instead adopt the Total Variation Distance (TVD) (Devroye et al., 1996) as a more robust metric:

$$\text{TVD}(P, Q) = \frac{1}{2} \sum_{i=1}^{N} |P_i - Q_i|, \tag{6}$$

where $P$ and $Q$ are probability distributions from the two models. The TVD is bounded in $[0, 1]$, making it numerically stable even for deterministic predictions, and maintains connections to information-theoretic measures through Pinsker's inequality (Pinsker, 1964).

**Measuring Predictive Multiplicity.** Having obtained a collection of models that satisfy the Rashomon criteria as defined in Eq. (2), we require metrics to characterize the extent and structure of predictive multiplicity. Unlike the metrics discussed in Section 2, these operate at the set level and summarize the properties of an entire collection of near-optimal solutions. We provide an overview of all the metrics used, their formal definitions, and a brief explanation in Appendix A.1.

*Decision-based Metrics.* One approach is to evaluate predictive multiplicity directly through model outputs. Ambiguity measures the proportion of instances whose predicted labels vary across models, while discrepancy captures the maximum proportion of disagreements between a baseline model and any alternative. Together, these metrics characterize whether multiplicity manifests as scattered disagreements or as systematic divergence across the Rashomon set (Marx et al., 2020).

*Probability-based Metrics.* Beyond decision-level measures, probability-based metrics capture how predictive multiplicity manifests in model output scores. The Viable Prediction Range (VPR) (Watson-Daniels et al., 2023) summarizes, for each sample, the range of predicted probabilities assigned across the Rashomon set, indicating how wide predictions can vary while maintaining accuracy. The Rashomon Capacity (RC) (Hsu & Calmon, 2022) instead measures score variations in the probability simplex using information-theoretic quantities, quantifying the effective number of distinct predictive distributions while avoiding the overestimation that can occur with thresholded decisions. Together, VPR highlights the spread of probabilities for a class, whereas RC captures the richness of distributional variation, providing complementary perspectives on multiplicity.

## 3 MODULATING ACTIVATIONS FOR RASHOMON SET GENERATION

Our method, DIVERSE, explores the Rashomon set of neural networks by embedding a pretrained model into a FiLM-modulated latent space and optimizing this space with CMA-ES. The approach unfolds in three steps: (i) training a reference model, (ii) defining a FiLM-modulated space of models, and (iii) exploring this space with CMA-ES under Rashomon constraints.

**Reference model.** We train a reference network $f_{\text{ref}}$ under standard supervised learning, with training stopped once validation performance stabilizes. The reference performance sets the Rashomon threshold, defined as the minimum performance required for inclusion in the set. All subsequent variants are initialized from $f_{\text{ref}}$'s learned weights.

**FiLM-modulated model space.** To induce controlled variation without retraining, we wrap the pretrained reference model with frozen FiLM layers. The reference latent vector is initialized to $z_{\text{ref}} = 0$, which recovers the unmodified network. Any nonzero choice of $z$ defines a candidate model $f_z$ that inherits all weights from $f_{\text{ref}}$ but differs in its internal activations.

We consider three different insertion strategies depending on the underlying architecture, as shown in Fig. 1. Regardless of the used strategy, FiLM acts on the pre-activations $h$, and all inserted FiLM layers share the same latent vector $z$. Concretely, this means that this single vector modulates the model in a coordinated, network-wide fashion.

Formally, FiLM applies a feature-wise affine transformation with modulation parameters ($\gamma$ and $\beta$),

$$FiLM(h; \boldsymbol{z}) = \gamma(\boldsymbol{z}) \odot h + \beta(\boldsymbol{z}), \quad \gamma(\boldsymbol{z}) = 1 + \tanh(\boldsymbol{z}W_\gamma), \quad \beta(\boldsymbol{z}) = \tanh(\boldsymbol{z}W_\beta), \quad (7)$$

where $W_\gamma, W_\beta \in \mathbb{R}^{d \times C}$ are frozen projection matrices with random initialization drawn from $\mathcal{N}(0, 0.5^2)$. The use of $\tanh$ bounds modulation to $\gamma \in [0, 2]$ and $\beta \in [-1, 1]$, preventing destabilizing amplification or large DC shifts during search. By construction, $\boldsymbol{z} = \boldsymbol{0}$ recovers the reference model ($\gamma = 1, \beta = 0$), while nonzero vectors induce structured, bounded variations. For dense outputs $h \in \mathbb{R}^{B \times C}$, FiLM acts feature-wise. For convolutional outputs $h \in \mathbb{R}^{B \times H \times W \times C}$, $\gamma, \beta \in \mathbb{R}^{B \times C}$ are reshaped to $(B, 1, 1, C)$ and broadcast across spatial dimensions, ensuring consistent per-channel modulation.

By freezing the projection matrices, the latent vector defines a reproducible, low-dimensional modulation space of functional variations. This design eliminates additional hyperparameters, making the method simple and stable while remaining expressive enough to generate diverse behaviors.

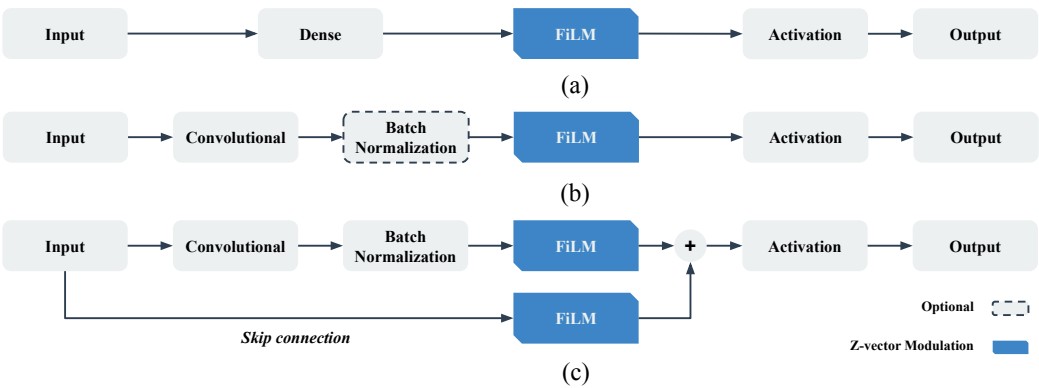

Figure 1: FiLM placement (blue): (a) after dense layers, (b) after convolutional blocks, after batch normalization when present, otherwise directly after the convolution, and (c) on residual skip connections. A shared latent vector $\boldsymbol{z}$ provides the modulation parameters for all FiLM layers per model.

**Latent Vector Design.** The latent vector $\boldsymbol{z} \in \mathbb{R}^d$ parameterizes the FiLM modulations defined in Eq. (7). This provides a modulation space: instead of operating in the full weight space of the network, CMA-ES explores a lower-dimensional latent space that indirectly controls network behavior. Its dimensionality $d$ determines the effective size of the search space explored by CMA-ES. Small $d$ values yield efficient exploration but could restrict the diversity of functional variants, since the same latent dimensions are projected across many features. Larger $d$ values could permit richer modulations and a broader range of behaviors, but increase computational cost and strain CMA-ES, which scales poorly in very high dimensions (Omidvar & Li, 2011).

Initialization also plays an important role. As previously discussed, $\boldsymbol{z} = \boldsymbol{0}$ recovers the pretrained reference model, making it a natural anchor point for search. However, because CMA-ES adapts its distribution locally, different starting means can bias exploration toward different regions of the modulation space, potentially uncovering distinct Rashomon members. To mitigate this sensitivity and study how initialization affects the Rashomon set that is uncovered, we evaluate multiple starting points, probing both near-identity and stronger modulation regimes (see Section 4).

**Searching the Latent Modulation Space with CMA-ES.** Given a choice of dimensionality and initialization (Section 3), exploration of the modulation space is performed with CMA-ES. At each generation, CMA-ES samples candidate latent vectors from a multivariate Gaussian distribution (Eq. (4)), evaluates their fitness, and then adapts the sampling distribution toward higher-scoring regions. This makes it well-suited for our setting: the search space is continuous, non-convex, and gradient information is unavailable since underlying network weights are frozen.

The key design choice is the fitness function, which must balance two objectives: (i) keeping candidates close to the reference model in performance, so they remain in the Rashomon set, and (ii) maximizing predictive disagreement with the reference, so that discovered models are diverse. To do this, we compute class probabilities on the training set $p_z(x)$, and form a loss- and diversity-aware objective. Let $p_{\text{ref}}(x)$ be the reference model's probabilities, and denote the cross-entropy

(CE) training loss of $f_{\boldsymbol{z}}$ and $f_{\text{ref}}$ by $L_{\text{train}}(\boldsymbol{z})$ and $L_{\text{train}}^{\text{ref}}$, respectively. We define the relative loss increase

$$\Delta(\boldsymbol{z}) = \frac{L_{\text{train}}(\boldsymbol{z}) - L_{\text{train}}^{\text{ref}}}{L_{\text{train}}^{\text{ref}} + 10^{-8}}, \tag{8}$$

where the denominator is stabilized with a small constant to prevent division by zero. Then we apply a Gaussian penalty centered at 0,

$$\phi_\epsilon(\boldsymbol{z}) = \exp(-\frac{\Delta(\boldsymbol{z})^2}{2\epsilon^2}), \tag{9}$$

which softly enforces the Rashomon parameter $\epsilon$: candidates with losses close to the reference (i.e., $\Delta(\boldsymbol{z}) \approx 0$) receive little penalty; larger deviations are exponentially down-weighted rather than hard-rejected. This choice avoids discarding potentially diverse candidates that lie near the Rashomon boundary, allowing CMA-ES to explore the trade-off between accuracy and disagreement more flexibly.

To capture functional diversity, we combine soft (Eq. (6)) and hard disagreement (Eq. (5)) with a mixing weight $\lambda \in [0, 1]$. This weighting allows us to tune their relative influence: when $\lambda = 0$, the score reflects only hard disagreement, while $\lambda = 1$ reduces it to purely soft disagreement. In our experiments, we fix $\lambda = 0.5$ to balance the contributions. A brief $\lambda$–ablation Appendix A.2 shows that results are mostly insensitive to this choice, due to the strong correlation between soft and hard disagreement in our FiLM-modulated models. The resulting diversity score is defined as:

$$\text{Div}_\lambda(\boldsymbol{z}) = \lambda\text{TVD}(p_{\text{ref}}(x), p_z(x)) + (1 - \lambda)\text{Dis}(f_{\text{ref}}, f_{\boldsymbol{z}}). \tag{10}$$

Finally, we combine the diversity score with the Gaussian penalty into a single fitness function:

$$\mathcal{F}(\boldsymbol{z}) = \text{Div}_\lambda(\boldsymbol{z}) \cdot \phi_\epsilon(\boldsymbol{z}). \tag{11}$$

Taken together, the fitness design steers CMA-ES towards models that remain within the Rashomon set while exhibiting meaningful disagreement at both the decision and probability levels. The Gaussian penalty softly preserves accuracy, preventing premature rejection of boundary cases, while the mixture of hard and soft disagreement captures complementary notions of diversity. To ensure these properties extend beyond the training distribution, we enforce the Rashomon constraint on the validation set, retaining only $\boldsymbol{z}$-candidates within $\epsilon$ of the reference model (Eq. (2)). The surviving models are then evaluated on a held-out test set, ensuring that reported Rashomon members reflect genuine diversity rather than overfitting to the search process.

## 4 EXPERIMENTAL SETUP

**Datasets and Reference Models.** We evaluate DIVERSE on three complementary benchmarks, each with an appropriate reference model. On MNIST (LeCun et al., 1998) (70k samples, 10 classes), we use a 3-layer MLP (98% val/test), testing applicability beyond convolutional architectures. On PneumoniaMNIST (Yang et al., 2023) (5.8k samples, 2 classes), we adopt an ImageNet-pretrained ResNet-50 (98% val, 91% test), reflecting standard transfer learning in medical imaging. On CIFAR-10 (Krizhevsky, 2009) (60k samples, 10 classes), we use an ImageNet-pretrained VGG-16 (77% val, 75% test), a common baseline for moderate-scale vision tasks. Reference models are trained with Adam and early stopping on validation performance; their accuracies define Rashomon thresholds and remain frozen during CMA-ES search. When no validation set is available, 10% of the training data is reserved. All subsequent variants are FiLM-modulated according to the insertion strategies in Section 3 and illustrated in Fig. 1.

**Z-vectors.** The latent vector $\boldsymbol{z} \in \mathbb{R}^d$ parameterizes FiLM modulations as described in Section 3. We study $d \in \{2, 4, 8, 16, 32, 64\}$, balancing search reach against CMA-ES scaling. As discussed in Section 3, initialization biases exploration toward different regions of the latent space. To probe this effect, for each $d$ we run 10 initializations: the natural anchor ($\boldsymbol{z} = \boldsymbol{0}$), a stronger perturbation ($\boldsymbol{z} = \boldsymbol{1}$), and eight additional draws from Gaussians centered at 0 or 1 ($\sigma = 0.1$). The same initialization seeds are reused across all experiments for comparability.

**CMA-ES Search Protocol.** We configure CMA-ES using full covariance settings, following CMA-ES defaults (Hansen, 2016), using a population size of popsize $= 4 + 3\log d$ for latent dimension $d$. To control the search horizon, we use a parameter $k$, which specifies the number of

evaluations we allocate per latent dimension. Setting $k = 80$ yields a target budget of $kd$ total evaluations, ensuring that the computational effort scales linearly with the size of the search space and that higher-dimensional $z$ receive proportionally more exploration. While CMA-ES has no strict rule for this choice, community practice often uses budgets of tens to hundreds of evaluations per dimension; our setting aligns with this and keeps computation tractable.

Because CMA-ES evaluates popsize candidates per generation, the evaluation budget $kd$ translates into $n_{\text{generations}} = \lceil kd / \text{popsize} \rceil$, which serves as the stopping criterion: the search runs until $n_{\text{generations}}$ full generations have been completed. Each generation produces popsize candidates, resulting in $m = \text{popsize} \cdot n_{\text{generations}}$ total models per run. Since both popsize and $n_{\text{generations}}$ depend on the latent dimension $d$, the total number of models $m$ generated by DIVERSE is fixed once $d$ is chosen. The ceiling operator may cause $m$ to exceed the nominal budget of $kd$ by at most one additional generation, but this remains negligible. Finally, to assess the sensitivity of DIVERSE to the choice of initial exploration scale, we ablate over initial step sizes $\sigma_0 \in \{0.1, 0.2, 0.3, 0.4, 0.5\}$.

**Metrics.** We evaluate the Rashomon sets along two dimensions: their size and their internal structure. The Rashomon Ratio (Section 2) quantifies the proportion of candidates that meet the Rashomon constraint, capturing test size. To assess structural diversity, we measure ambiguity and discrepancy to capture decision-level variation, and employ the Variable Prediction Range (reported as the width of the range rather than its bounds) and Rashomon Capacity (Section 2) to characterize distributional variation, with the latter reflecting the number of distinct predictive distributions in the set. Together, these metrics provide a comprehensive characterization of multiplicity within the discovered Rashomon sets.

**Baselines.** We compare DIVERSE against two families of baselines. First, retraining-based, obtained by reinitializing and retraining the reference model with different seeds, represents the standard but computationally costly approach to Rashomon exploration. Second, we include dropout-based Rashomon exploration (Hsu et al., 2024), utilizing the Gaussian sampling method introduced by them, which implicitly samples functional variations by applying dropout masks at evaluation time. Unlike our method, however, dropout defines Rashomon membership directly on the test set, which risks optimistic estimates since the same data is used for both selection and evaluation. Our protocol instead enforces Rashomon constraints on a validation set, and reports diversity metrics only on held-out test data, yielding stricter but unbiased comparisons. Because of the high computational cost and impracticality of AWP for large neural networks, as discussed in Section 2, we do not include Adversarial Weight Perturbation (Hsu & Calmon, 2022) as a baseline. For baseline comparisons, we report DIVERSE under the best-performing hyperparameter configuration per $\epsilon$, selected by averaging performance across all multiplicity metrics; the corresponding configurations are listed in Table 3. All methods were executed on the same hardware; full details are in Appendix A.3.

## 5 RESULTS

We evaluate DIVERSE against retraining and dropout baselines on MNIST, PneumoniaMNIST, and CIFAR-10. Our analysis addresses three questions: (i) how large and diverse are the Rashomon sets discovered, (ii) how robust is the method to hyperparameters, and (iii) how does DIVERSE compare to the baselines.

**Rashomon Sets and Latent Dimension.** Fig. 2 reports five metrics across Rashomon thresholds $\epsilon$: Rashomon Ratio, discrepancy, ambiguity, Viable Prediction Range (VPR), and Rashomon Capacity. On MNIST, nearly all modulated models satisfy the tolerance (ratios $\approx 1.0$), while discrepancy and ambiguity grow with $\epsilon$, showing that DIVERSE uncovers functionally varied solutions. PneumoniaMNIST and CIFAR-10 yield lower ratios, reflecting the challenge of preserving performance in deeper models, yet diversity metrics still increase with $\epsilon$, confirming meaningful multiplicity.

Latent dimension $d$ further shapes these outcomes. On MNIST, all combinations of $\epsilon$ and $d$ yield Rashomon sets, with larger $d$ providing only small gains in diversity. At $d = 32$ and $d = 64$, the Rashomon Ratio shows jitter at the strictest thresholds ($\epsilon = \{0.01, 0.02\}$). On CIFAR-10 and PneumoniaMNIST, the effect is more restrictive: $d \in \{2, 4\}$ yields sets across all $\epsilon$, $d = 8$ only at $\epsilon \geq 0.03$, and no sets are recovered for $d \geq 16$.

**Robustness to Hyperparameters.** We evaluated sensitivity to the CMA-ES step size $\sigma_0$ (Fig. 6) and latent initialization (Fig. 7). Initializing the latent vector at $z = 0$ consistently produced

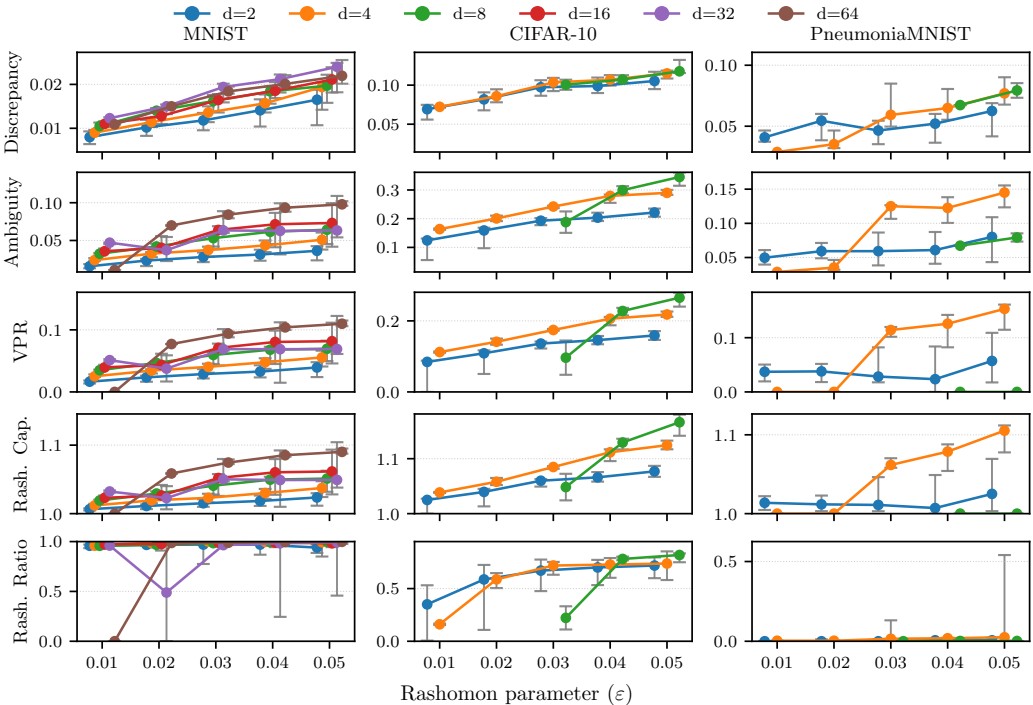

Figure 2: Diversity metrics (discrepancy, ambiguity, Rashomon capacity, VPR) and Rashomon Ratio across Rashomon thresholds ($\epsilon$) for varying latent dimensions ($d$). Each group of points corresponds to one $\epsilon$; horizontal spacing within groups is only for readability and carries no semantic meaning. Markers show medians, error bars the interquartile range (IQR) across latent initializations, CMA-ES step sizes ($\sigma_0$), and $d$.

Rashomon members across datasets, whereas initialization around $z = 1$ often failed. An exception is MNIST, where such initializations still yielded Rashomon members, though with reduced Rashomon ratios. The effect of $\sigma_0$ also varies across datasets: larger initial step sizes increase diversity on MNIST, while smaller step sizes are more effective on PneumoniaMNIST and CIFAR-10.

**Baseline Comparison and Runtime Efficiency.** Fig. 4 compares DIVERSE, dropout-based sampling, and retraining across diversity metrics and the Rashomon Ratio. Retraining achieves the highest diversity metrics in most cases, but does not always yield the largest Rashomon Ratios. On MNIST, DIVERSE exceeds retraining on discrepancy at higher $\epsilon$ values and outperforms dropout across all metrics except VPR. On CIFAR-10, DIVERSE surpasses dropout across all metrics. On PneumoniaMNIST, DIVERSE outperforms dropout and retraining on discrepancy at the highest $\epsilon$-level, but falls short at stricter thresholds. Across datasets, DIVERSE attains consistently lower VPR values. Table 1 compares the runtimes to generate $m$ candidates. Retraining is prohibitively costly, requiring hours on PneumoniaMNIST and CIFAR-10, dropout samples models within seconds, and DIVERSE completes search in minutes, but remains slower than dropout-based sampling.

**Qualitative Illustrations of Multiplicity** To complement the quantitative metrics, we provide a qualitative view of predictive multiplicity in Fig. 3. For each dataset (MNIST, CIFAR-10, and PneumoniaMNIST), we select the sample with the highest model disagreement and display the input alongside the class-frequency histogram over the Rashomon set found by DIVERSE. These examples reveal how models that are all equally optimal under the Rashomon constraint can nevertheless make different decisions on the same input, making predictive multiplicity visually explicit.

**Layerwise Sensitivity.** We additionally performed a layerwise $\Delta$TVD sensitivity analysis to identify which FiLM layers contribute most to disagreement (details in Appendix A.5). Across architectures, diversity is localized: early FiLMs dominate in MNIST, mid-level convolutional FiLMs in VGG16, and early to mid-stage FiLMs in ResNet50. These patterns suggest that only a subset

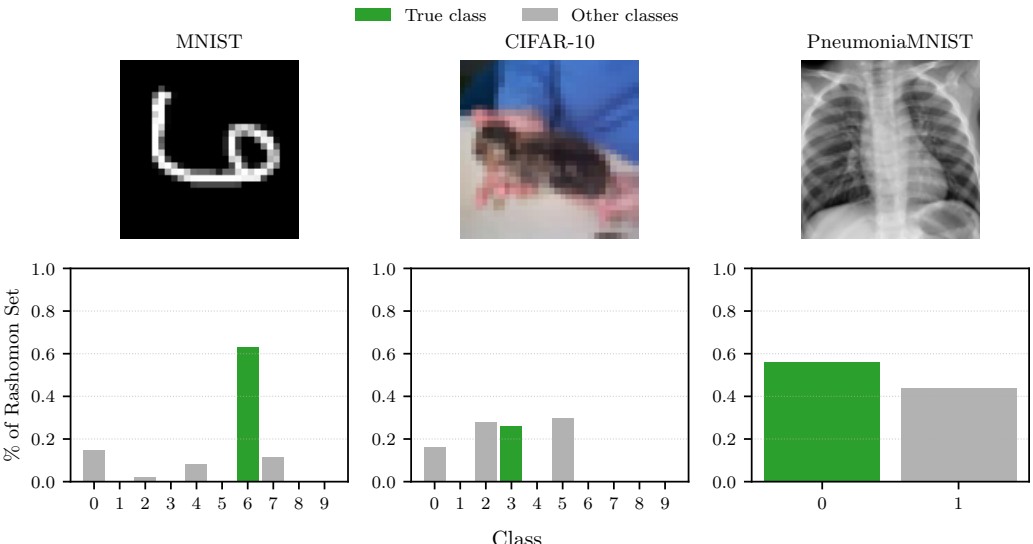

Figure 3: Highest-disagreement samples for MNIST, CIFAR-10, and PneumoniaMNIST. Each panel displays the input image and the class-frequency distribution over the Rashomon set members; the true class is shown in green, and all other classes are represented in gray.

of FiLM sites meaningfully drives diversity, pointing toward future approaches that focus CMA-ES search on sensitivity-identified layers rather than modulating the entire network.

Taken together, these results demonstrate that DIVERSE consistently generates non-trivial Rashomon sets under principled validation constraints, with efficiency that enables exploration at scale. We next discuss implications for multiplicity research and limitations of our approach.

Table 1: Runtime (hh:mm:ss) to obtain $m$ candidate models under the same Rashomon constraint. For DIVERSE, $m$ is determined by the latent dimension $d$ (see Section 4); in the settings shown, $d = 2$ yields $m = 162$ models and $d = 8$ yields $m = 640$. All baselines are configured to generate the same $m$ for fair comparison.

| Method | MNIST | | PneumoniaMNIST | | CIFAR-10 | |
|---|---|---|---|---|---|---|
| | $m=162$ | $m=640$ | $m=162$ | $m=640$ | $m=162$ | $m=640$ |
| Retrain | 00:29:39 | 01:57:36 | 02:09:00 | 08:16:00 | 03:17:26 | 12:37:27 |
| Dropout | 00:00:30 | 00:01:57 | 00:01:32 | 00:05:47 | 00:01:58 | 00:06:30 |
| **DIVERSE** | **00:00:50** | **00:03:16** | **00:01:49** | **00:07:15** | **00:02:11** | **00:08:42** |

## 6 DISCUSSION

**Key Findings.** Our study shows that DIVERSE reliably uncovers non-trivial Rashomon sets across varying datasets. It recovers functionally distinct models under strict $\epsilon$, is robust to initialization and step size settings, and produces candidate sets within minutes. Compared to existing approaches, DIVERSE is far more efficient than retraining while delivering equal or better diversity than dropout under principled validation and test constraints. These findings establish DIVERSE as a practical tool for large-scale multiplicity research in deep neural networks.

**Interpretation of Results.** The experiments reveal how dataset complexity and architecture influence which configurations yield valid Rashomon sets. On MNIST, Rashomon members appear for all combinations of $\epsilon$ and latent dimension $d$, though diversity gains decline at larger $d$ and the Rashomon Ratio exhibits jitter at very small $\epsilon$ values. This suggests boundary effects under tight tolerances, where small fluctuations in validation performance determine membership. By con-

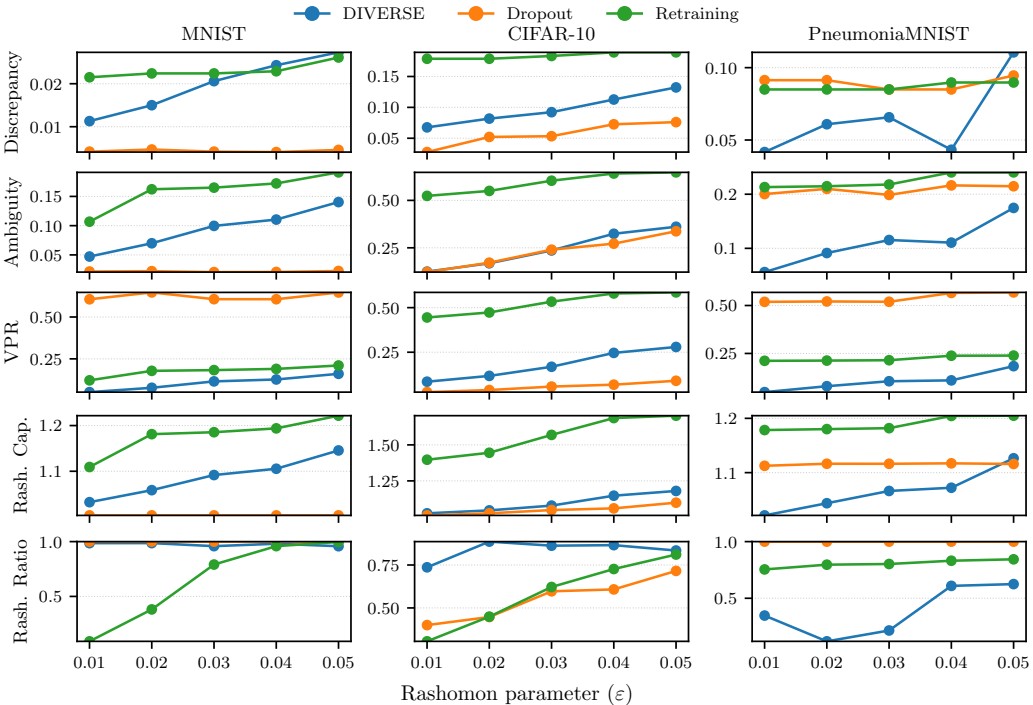

Figure 4: Best hyperparameter configuration for DIVERSE compared against the pre-defined parameters of the dropout authors vs retraining.

trast, PneumoniaMNIST and CIFAR-10 admit Rashomon sets only at small latent dimensions, with $d \geq 16$ failing entirely. This points to the more complex search landscapes created by deeper networks with many FiLM injections, where increased dimensionality makes it harder for CMA-ES to locate candidates within the Rashomon tolerance. Hyperparameter sensitivity follows the same pattern: larger initial step sizes encourage exploration in the simpler MNIST MLP, whereas smaller values are more effective for deeper models, where rugged landscapes benefit from finer search granularity. These results also contextualize comparisons with baselines: retraining provides broad exploration at high cost, while dropout is nearly instantaneous but bypasses search on the training set and defines Rashomon membership directly on the test set, leading to diversity estimates that may be optimistically biased by model selection on test data. Taken together, these findings suggest that multiplicity in deep networks is shaped not only by optimization but by the interplay of search dimensionality, architectural complexity, tolerance thresholds, and the evaluation protocol itself.

**Extension to Transformers.** While our main experiments focus on MLPs and CNNs, the FiLM-based modulation in DIVERSE also applies to dense layers in Vision Transformers (ViT) (Dosovitskiy et al., 2021). We include a small preliminary ViT experiment on CIFAR-10 in Appendix A.6, which provides an initial indication that the modulation–search mechanism extends to Transformer architectures and motivates a more comprehensive future study.

**Limitations.** While DIVERSE shows that FiLM-based modulation with CMA-ES can efficiently uncover non-trivial Rashomon sets across multiple architectures and domains, several opportunities remain for improvement. First, full-covariance CMA-ES becomes costly as latent dimensionality grows, restricting us to moderate-sized modulation spaces; scalable variants such as DD-CMA-ES (Akimoto & Hansen, 2019), offer a promising path toward higher-dimensional search. Second, FiLM modulation provides controlled functional variation but does not guarantee that the discovered models differ in interpretable or causally meaningful ways. Third, although our evaluation spans three datasets, extending to additional modalities, including a more extensive Vision Transformer study (Appendix A.6), would further test the generality of our conclusions.

REPRODUCIBILITY STATEMENT

We have taken several measures to ensure the reproducibility of our work. All datasets used (MNIST, CIFAR-10, and PneumoniaMNIST) are publicly available, with preprocessing and splits described in Section 4. Detailed descriptions of the methodology, including FiLM modulation, CMA-ES search, and the fitness function, are provided in Section 3, with full metric definitions in Appendix A.1. Hyperparameter settings and search protocols are outlined in Section 4, and additional sensitivity analyses are reported in Appendix A.5. A complete specification of the hardware setup is provided in Appendix A.3. To facilitate the exact reproduction of our experiments, we release the full source code and experiment scripts at `https://github.com/UHasselt-DigitalFutureLab/Diverse`.

ETHICS STATEMENT

This work focuses on methods for exploring the Rashomon set of neural networks. All datasets used (MNIST, CIFAR-10, and PneumoniaMNIST) are publicly available and widely adopted for academic research under appropriate licenses. No personally identifiable information or sensitive private data were collected or generated in this study. We have taken care to ensure fairness by reporting results across multiple datasets and by using a validation-based Rashomon membership criterion to avoid biased or overly optimistic evaluations. The potential risks of this research are limited to the misuse of diverse model sets for generating deceptive predictions; however, we emphasize that our intended contribution is to enhance the robustness, transparency, and understanding of predictive multiplicity.

ACKNOWLEDGMENTS

This work was funded by the Flemish Government under the "Onderzoeksprogramma Artificiële Intelligentie (AI) Vlaanderen" programme, R-13509. This research was partly funded by the Special Research Fund (BOF) of Hasselt University (R-14436) and the FWO fellowship grant (1SHDZ24N). The resources and services used in this work were partly provided by the VSC (Flemish Supercomputer Center), funded by the Research Foundation - Flanders (FWO) and the Flemish Government.

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

## A APPENDIX

### A.1 PREDICTIVE MULTIPLICITY METRICS.

We begin by introducing the notation used throughout this appendix. Let $\mathcal{D} = \{(x_i, y_i)\}_{i=1}^N$ denote the dataset with $N$ samples. Let $f_{\text{ref}}$ be the reference model obtained under standard training. The empirical Rashomon set, constructed according to Eq. (2), is given by

$$\mathcal{R}_\epsilon^m = \{f^{(1)}, f^{(2)}, \dots, f^{(M)}\},$$

where each $f^{(j)}$ is a candidate model within the tolerance $\epsilon$. For a model $f^{(j)} \in \mathcal{R}_\epsilon^m$, we denote its predicted label on input $x_i$ by

$$\hat{y}_i^j = f^{(j)}(x_i),$$

and the prediction of the reference model by

$$\hat{y}_i^{\text{ref}} = f_{\text{ref}}(x_i).$$

**Rashomon Ratio.** As introduced by Semenova et al. (2022), the Rashomon Ratio measures the relative size of a Rashomon set. Adapting this definition to the empirical Rashomon set $\mathcal{R}_\epsilon^m$, it is given by:

$$\text{RR}_\epsilon = \frac{|\mathcal{R}_m^\epsilon|}{|\mathcal{H}_m|}. \tag{A.1}$$

**Ambiguity.** Following Marx et al. (2020), the fraction of samples for which at least one Rashomon model disagrees with the reference prediction:

$$\text{Ambiguity}(\mathcal{R}_\epsilon^m, f_{\text{ref}}) = \frac{1}{N} \sum_{i=1}^N \mathbb{1}\left[\exists j \in \{1, \dots, M\} : \hat{y}_i^j \neq \hat{y}_i^{\text{ref}}\right] \tag{A.2}$$

**Discrepancy.** Following Marx et al. (2020), the worst-case disagreement rate between the reference model and any Rashomon model:

$$\text{Discrepancy}(\mathcal{R}_\epsilon^m, f_{\text{ref}}) = \max_{j \in \{1, \dots, M\}} \frac{1}{N} \sum_{i=1}^N \mathbb{1}\left[\hat{y}_i^j \neq \hat{y}_i^{\text{ref}}\right], \tag{A.3}$$

**Viable Prediction Range (VPR).** As introduced by Watson-Daniels et al. (2023), for a sample $x_i$, model $f^{(j)} \in \mathcal{R}_\epsilon^m$ outputs a class–probability vector $p_{i,\cdot}^{(j)} \in \Delta^{k-1}$. For class $c \in \{1, \dots, k\}$, define the per-class VPR as

$$\text{VPR}_{i,c} = \max_{1 \le j \le M} p_{i,c}^{(j)} - \min_{1 \le j \le M} p_{i,c}^{(j)}. \tag{A.4}$$

In practice we report the true-class width $\text{VPR}_{i,y_i}$ and summarize $\{\text{VPR}_{i,y_i}\}_{i=1}^N$ over the test set.

**Rashomon Capacity (RC).** Proposed by Hsu & Calmon (2022) and empirically computed as in Hsu et al. (2024), the RC measures the effective number of distinct predictive distributions in the set: we fix $x_i$ and view the empirical Rashomon set as a discrete channel with input $W \in \{1, \dots, M\}$ (model index) and output $Y \in \{1, \dots, k\}$ (predicted class), where $P(Y = c \mid W = j) = p_{i,c}^{(j)}$. The per-sample Rashomon Capacity is the channel capacity:

$$\text{RC}_i = \max_{r \in \Delta_M} I(W; Y) = \max_{r \in \Delta_M} \sum_{j=1}^M r_j \, D_{\text{KL}}\left(p_{i,\cdot}^{(j)} \,\Big\|\, \sum_{\ell=1}^M r_\ell \, p_{i,\cdot}^{(\ell)}\right), \quad \text{(bits)}. \tag{A.5}$$

We compute $\text{RC}_i$ via the Blahut–Arimoto algorithm (using $\log_2$). Set-level summaries (mean/median) are reported over $\{\text{RC}_i\}_{i=1}^N$.

## A.2 IMPACT OF MIXING WEIGHT

To assess the influence of the mixing weight $\lambda$ in Eq. (10), which balances soft (TVD-based) and hard (label-based) disagreement, we conducted a controlled ablation on all three datasets. We fixed all other hyperparameters to ($d = 2$, $z_0 = $ z0zeros, $\epsilon = 0.05$) and varied $\lambda \in \{0.0, 0.25, 0.50, 0.75, 1.0\}$. This setup isolates the effect of $\lambda$ by ensuring that it is the only variable that changes.

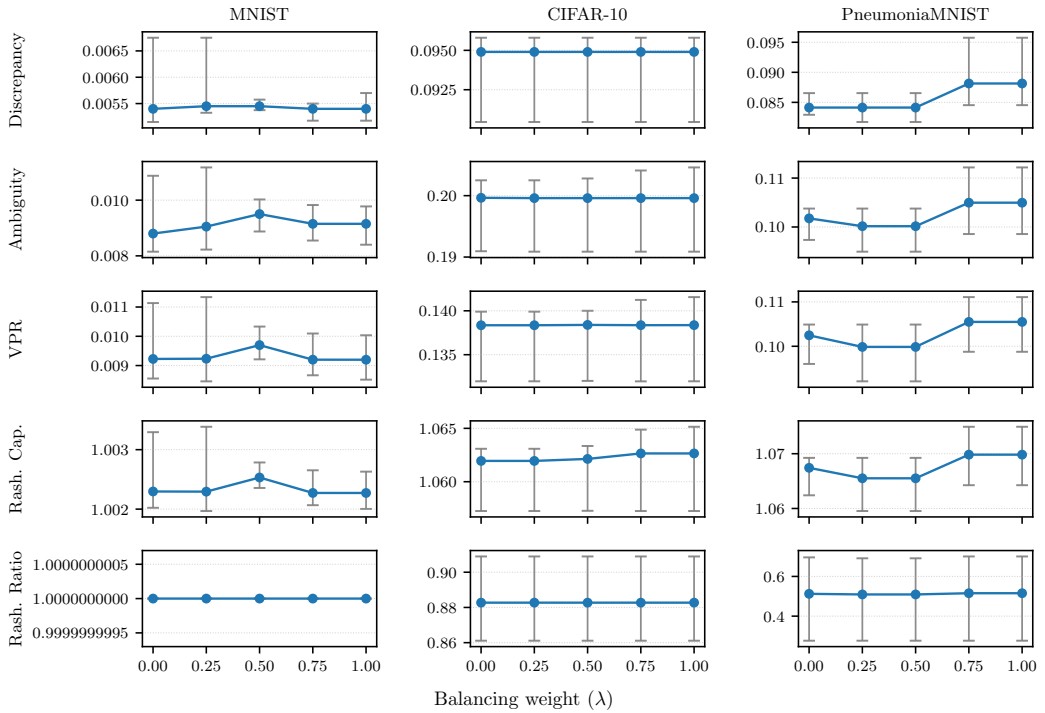

Figure 5: Ablation over the mixing weight $\lambda$. Across all datasets, diversity and Rashomon-set metrics change only marginally over $\lambda \in 0.0, 0.25, 0.5, 0.75, 1.0$. Median curves and IQR bands are computed over different $z_0$ initializations of the FiLM latent vector.

As shown in Fig. 5, the $\lambda$-ablation reveals that diversity metrics remain nearly invariant for most settings, yet a closer cross-dataset comparison uncovers the beginnings of an architectural trend. Our three benchmarks form a natural complexity hierarchy: MNIST (a shallow 3-layer MLP), CIFAR-10 (VGG-16), and PneumoniaMNIST (ResNet-50). Recall that in the diversity score of Eq. (10), setting $\lambda = 0$ makes the score depend only on hard disagreement, while $\lambda = 1$ makes it depend entirely on TVD-based soft disagreement; this score is used directly by CMA-ES to rank candidate latent vectors at each generation. We observe that as model complexity increases, soft (TVD-based) and hard (label-based) disagreement begin to decouple. This effect is most visible in PneumoniaMNIST, where higher $\lambda$ values—which place full weight on TVD in the diversity score—produce a small but noticeable increase in diversity. In contrast, MNIST shows virtually no $\lambda$-dependence, consistent with its highly overdetermined and near-deterministic predictions. These results suggest that deeper, higher-capacity architectures support richer probability-level variation that does not immediately translate into label flips, making soft disagreement increasingly informative. Consequently, $\lambda$ may become a more influential hyperparameter for Rashomon-set exploration in more expressive models, motivating future studies on wider CNNs and Transformer-based architectures.

## A.3 HARDWARE SETUP.

All experiments were run on a workstation equipped with an NVIDIA GeForce RTX 4090 GPU (24 GB VRAM), an Intel Core i9-14900K CPU, and 64 GB of system memory.

## A.4 IMPACT OF INITIAL STEP SIZE AND Z INITIALIZATION

We analyzed how CMA-ES hyperparameters influence Rashomon recovery. Figure 6 shows that the initial step size $\sigma_0$ affects the balance between exploration and precision: larger values promote greater diversity on simpler architectures such as MNIST, while smaller values are more effective on deeper networks, where fine-grained search helps maintain Rashomon membership. Figure 7 illustrates the effect of latent vector initialization. Starting from the anchor $z = 0$ consistently yields Rashomon members across datasets, while initializations centered around stronger perturbations ($z = 1$) often fail to recover candidates, except on MNIST. These results confirm that both hyperparameters shape the diversity and robustness of the discovered sets, underscoring the importance of initialization when exploring the modulation space.

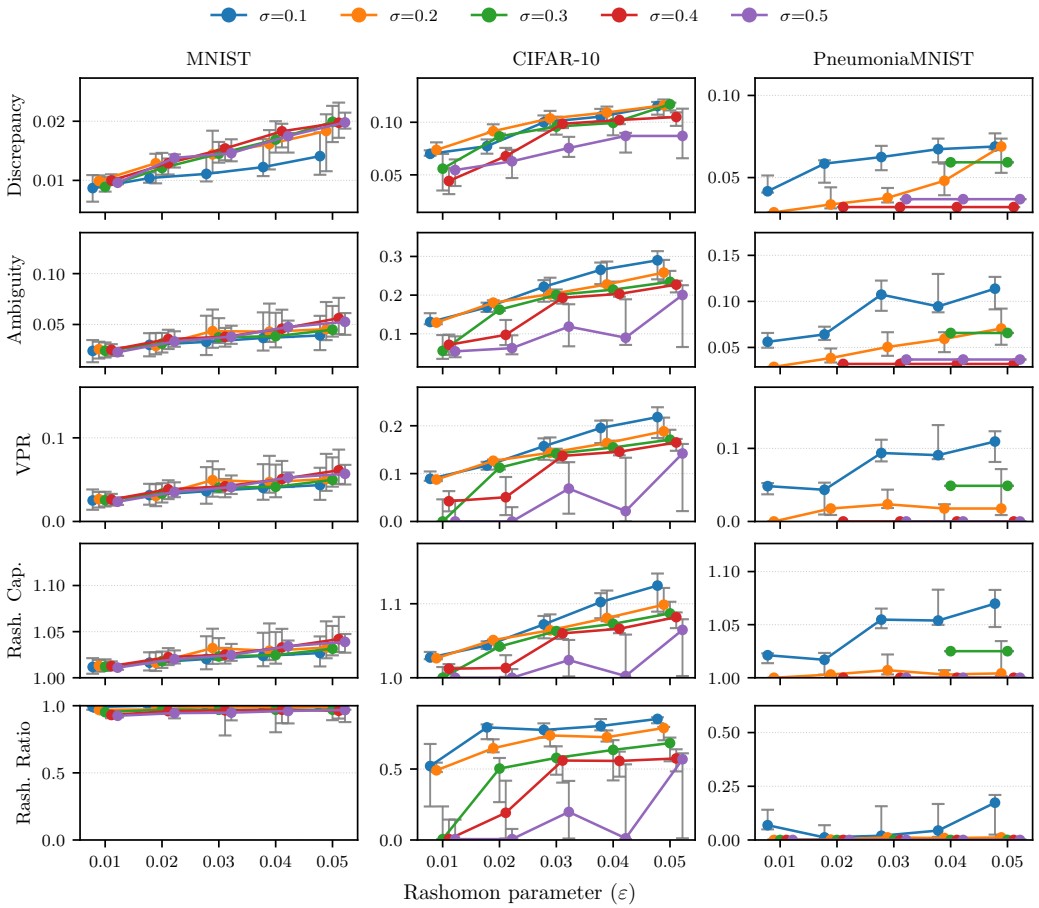

Figure 6: Diversity metrics (discrepancy, ambiguity, Rashomon capacity, and Viable Prediction Range) and Rashomon Ratio across Rashomon thresholds ($\epsilon$) for varying step sizes ($\sigma_0$). The x-axis values are discrete: each group corresponds to the same $\epsilon$ level, and the horizontal spacing within groups is for readability only and carries no semantic meaning. Markers denote the median across runs; error bars show the interquartile ranges (IQR), reflecting variability across latent vector initializations and CMA-ES initial step sizes ($\sigma_0$) and latent dimensions ($d$).

## A.5 LAYERWISE SENSITIVITY STUDY

To determine which FiLM layers contribute most to predictive disagreement, we performed a layerwise $\Delta$TVD sensitivity analysis on the best-performing Rashomon set obtained for each dataset. Because each FiLM layer in our reconstructed models is assigned an explicit, architecture-specific name (e.g., `film_1`, `film_conv_7`, `film_res_12`), we can unambiguously control, disable, and interpret individual modulation sites. These FiLM layers correspond to distinct functional locations

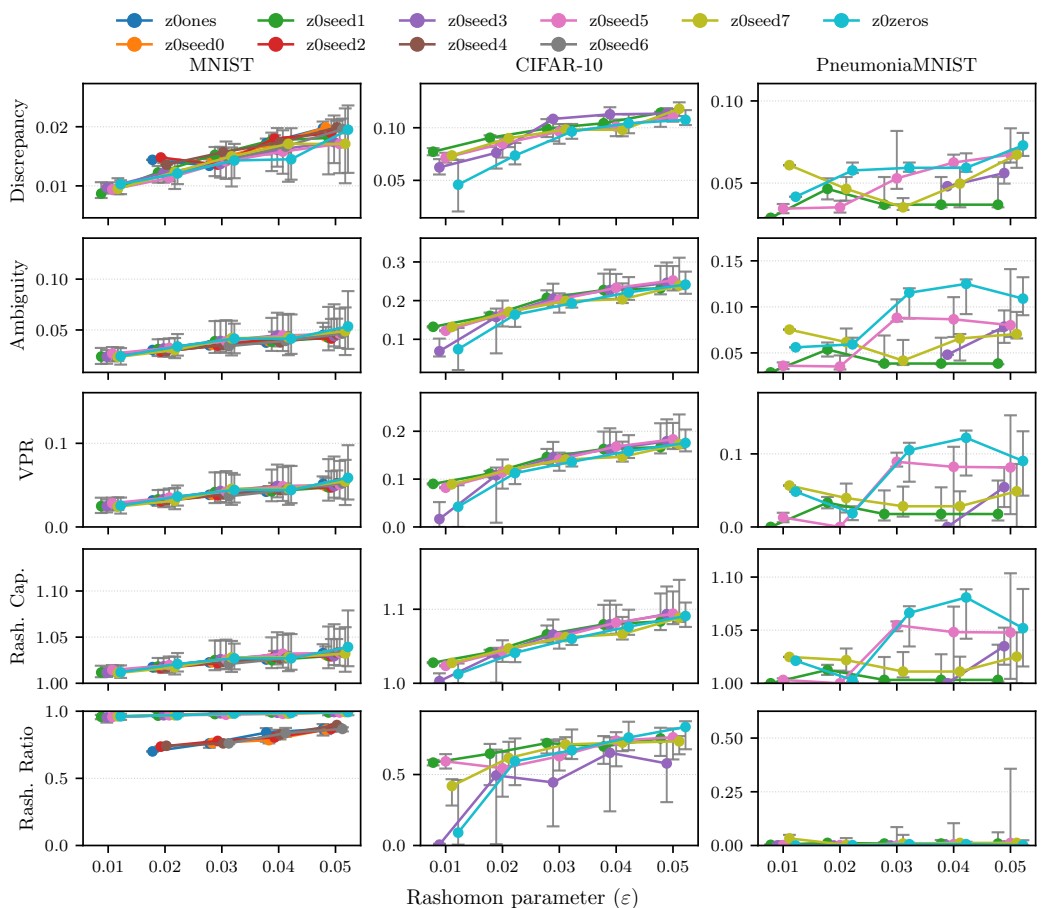

Figure 7: Diversity metrics (discrepancy, ambiguity, Rashomon capacity, and Viable Prediction Range) and Rashomon Ratio across Rashomon thresholds ($\epsilon$) for varying latent vector initializations. The seeds sampled around 0 are: z0seed1, z0seed3, z0seed5, z0seed7, while: z0seed0, z0seed2, z0seed4 and z0seed6 are sampled around 1. The x-axis values are discrete: each group corresponds to the same $\epsilon$ level, and the horizontal spacing within groups is for readability only and carries no semantic meaning. Markers denote the median across runs; error bars show the interquartile ranges (IQR), reflecting variability across latent vector initializations and CMA-ES initial step sizes ($\sigma_0$) and latent dimensions ($d$).

in the base architecture: modulation of dense-layer activations in the MNIST MLP, modulation before ReLU after each convolution in VGG16, and modulation between BatchNorm and ReLU or along residual branches in ResNet50. This naming scheme ensures that the effect of turning off a particular FiLM site has a clear architectural interpretation.

After CMA-ES identifies a Rashomon set using all FiLM layers, we take each member's latent vector $z$ and measure its baseline disagreement with the reference model (all FiLM layers active) using Total Variation Distance (TVD, Eq. (6)). We then disable one FiLM layer at a time, recompute predictions, and record the resulting change in disagreement. A positive $\Delta$TVD indicates that the removed FiLM layer resulted in more disagreement (its modulation increased diversity), whereas a negative value suggests the opposite

$$\Delta\text{TVD} = \text{TVD}_{\text{all}-\text{FiLM}-\text{on}} - \text{TVD}_{\text{layer}-\text{off}} \tag{12}$$

Since all FiLM layers share the same latent vector and initialization, differences in $\Delta$TVD arise solely from the architectural location of each FiLM site, making the sensitivity profile directly in-

terpretable. Averaging ΔTVD across all Rashomon members aggregates the contributions of many different latent vectors, meaning that different models may rely on different FiLM layers to generate disagreement. Thus, the reported values capture the average marginal influence of each FiLM layer across the Rashomon set. This analysis does not account for potential interactions between FiLM layers (e.g., the joint effect of disabling layers 1 and 3), which we leave as an interesting direction for future work.

Across architectures, disagreement is localized: early FiLM layers dominate in MNIST, mid-level convolutional FiLMs contribute most in CIFAR10, and early to mid-stage FiLMs dominate in PneumoniaMNIST. Several deeper FiLM sites yield negative ΔTVD, indicating that late-stage modulations often stabilize rather than diversify predictions.

These findings provide a structural interpretation of how FiLM layers influence multiplicity and suggest a promising direction for future work: using ΔTVD sensitivity scores to automatically select FiLM sites, allowing DIVERSE to focus CMA-ES search on layers that meaningfully affect functional diversity.

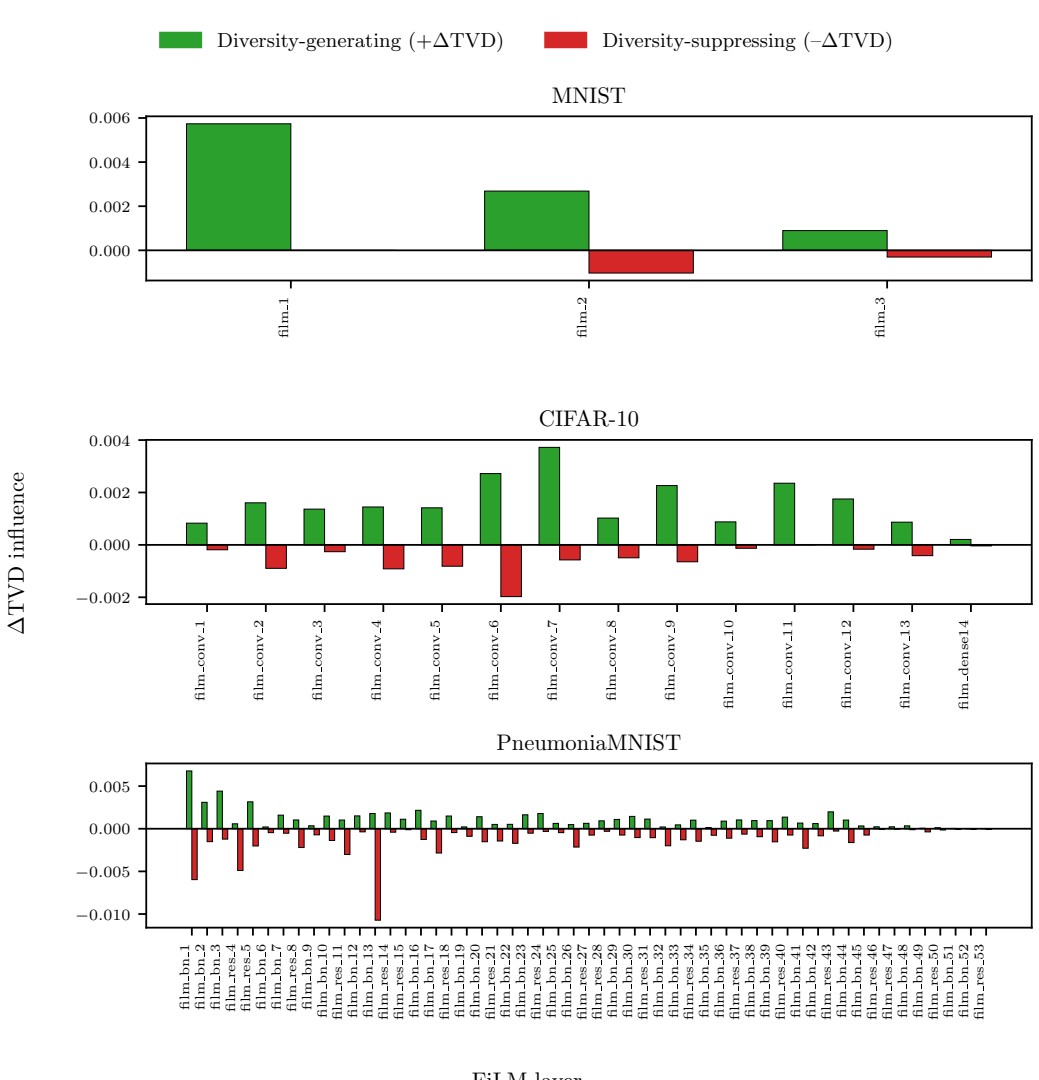

Figure 8: Layerwise ΔTVD sensitivity across MNIST, CIFAR10, and PneumoniaMNIST. Positive bars (green) indicate FiLM layers whose removal reduces disagreement (diversity-generating), while negative bars (red) indicate layers whose removal increases agreement (diversity-suppressing).

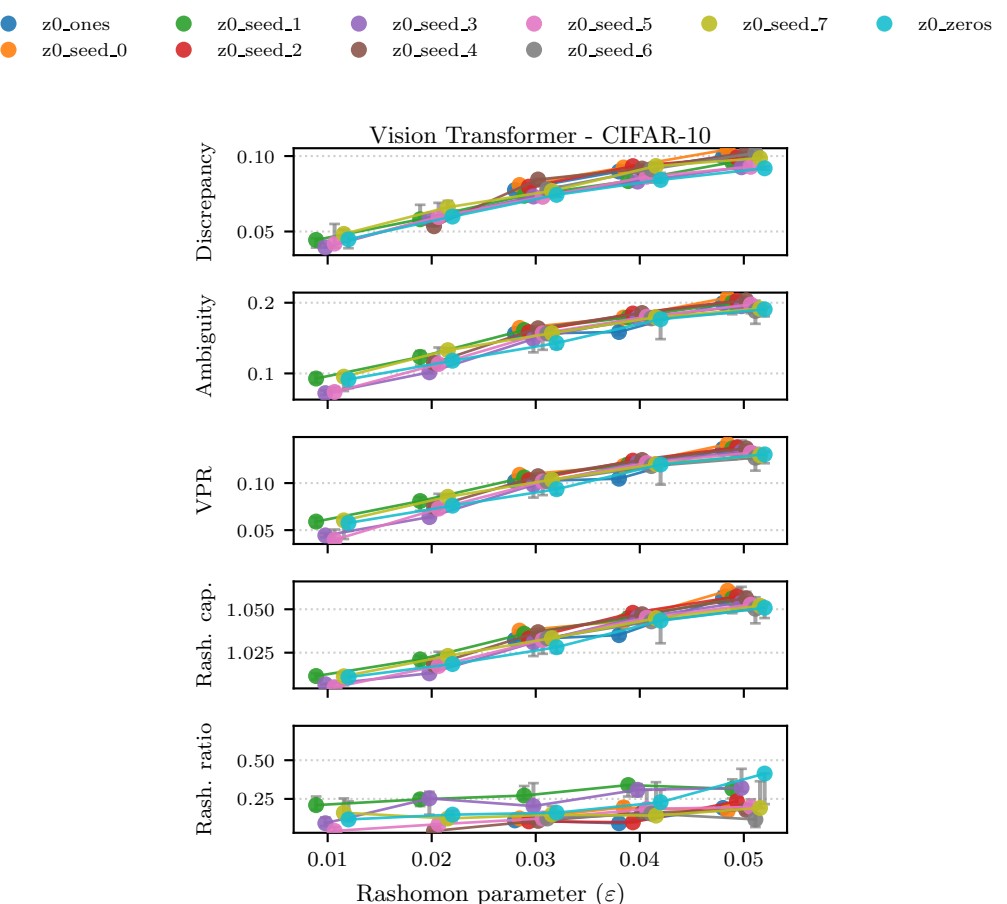

Figure 9: Diversity metrics and Rashomon Ratio for the Vision Transformer across Rashomon thresholds ($\epsilon$), grouped by latent vector initialization. Each point shows the median; error bars indicate IQR across runs. Horizontal spacing within each $\epsilon$ group is for readability only.

## A.6 APPLICABILITY ON VISION TRANSFORMERS

To further assess whether DIVERSE generalizes beyond convolutional and MLP-based architectures, we conducted an additional experiment on a Vision Transformer (ViT) (Dosovitskiy et al., 2021). This test evaluates whether a transformer, with deep stacking, multi-head self-attention, and high-dimensional MLP blocks, can also be embedded into a FiLM-modulated latent space that CMA-ES can effectively explore.

For this experiment, we use a compact ViT trained on CIFAR-10. Images are resized to 72×72, split into small patches, and embedded into a low-dimensional sequence representation. The encoder consists of 8 transformer blocks with 4-head self-attention, followed by a lightweight MLP classification head. The reference ViT is trained with standard supervised learning using AdamW, a batch size of 256, and up to 75 epochs, achieving 73% test accuracy before FiLM modulation.

To apply DIVERSE, we wrap the ViT with FiLM layers following the same dense-layer insertion rule shown in Fig. 1. Concretely, every dense layer inside the eight transformer MLP blocks, as well as the dense layers in the classification head, is FiLM-modulated. Attention layers, patch embeddings, and positional encodings remain unchanged in this study, though modulating these components presents an interesting direction for future work. As in all other architectures, every FiLM layer shares the same latent vector $z \in \mathbb{R}^d$, defined in Eq. (7). CMA-ES is run exactly as described in Section 4. For this initial test, we evaluate only $d = 2$, as the goal is to verify architectural generality rather than perform an exhaustive hyperparameter study.

All ViT runtime measurements (reported in Table 2) use this same configuration, with initial step size $\sigma_0 = 0.1$, Rashomon tolerance $\epsilon = 0.05$, and initialization at z0_zeros. The resulting timings are broadly comparable to those of VGG16 reported in Table 1, with the ViT being slightly slower, possibly due to its deeper sequence-processing architecture.

Across different $z_0$ initializations, the resulting Rashomon sets exhibit diversity metrics comparable to those obtained with VGG16 with $d = 2$ (Fig. 9). However, both the diversity outcomes and the timing measurements should be regarded as preliminary: more extensive experimentation is needed to fully validate performance on transformer-based models, including exploring larger latent dimensions and selectively modulating only the FiLM layers that meaningfully influence diversity (as suggested by the sensitivity analysis in Appendix A.5). Nonetheless, this experiment provides encouraging evidence that DIVERSE can operate on transformer-based architectures as well.

Table 2: Runtime (hh:mm:ss) to obtain $m$ candidate models for the Vision Transformer under the same Rashomon constraint.

| Method | Vision Transformer | |
|---|---|---|
| | $m{=}162$ | $m{=}640$ |
| **DIVERSE** | **00:02:16** | **00:08:52** |

### A.7 BEST PERFORMING HYPERPARAMETER FOR BASELINE COMPARISON

The hyperparameter settings used for comparing DIVERSE against the retraining and dropout baselines are reported in Table 3. For each dataset and Rashomon tolerance $\epsilon$, we select the configuration that achieves the best average performance across all multiplicity metrics: discrepancy, ambiguity, viable prediction range (VPR), Rashomon Capacity (RC), and Rashomon Ratio. The hyperparameters considered are the latent modulation dimension $d$, the CMA-ES initial step size $\sigma_0$, and the latent initialization $z_0$.

Table 3: Best performing hyperparameter settings for each dataset across the five Rashomon tolerances $\epsilon$.

| $\epsilon$ | MNIST | | | CIFAR-10 (VGG) | | | PneumoniaMNIST (ResNet-50) | | |
|---|---|---|---|---|---|---|---|---|---|
| | $d$ | $\sigma_0$ | $z_0$ | $d$ | $\sigma_0$ | $z_0$ | $d$ | $\sigma_0$ | $z_0$ |
| 0.05 | 64 | 0.2 | z0_zeros | 8 | 0.1 | z0_seed_5 | 4 | 0.1 | z0_seed_5 |
| 0.04 | 32 | 0.2 | z0_seed_5 | 8 | 01 | z0_zeros | 4 | 0.2 | z0_seed_5 |
| 0.03 | 32 | 0.2 | z0_seed_7 | 4 | 0.1 | z0_zeros | 2 | 0.1 | z0_zeros |
| 0.02 | 64 | 0.1 | z0_zeros | 2 | 0.1 | z0_zeros | 2 | 0.1 | z0_seed_7 |
| 0.01 | 32 | 0.1 | z0_zeros | 2 | 0.1 | z0_zeros | 2 | 0.1 | z0_zeros |

### A.8 LLM USAGE

Large language models (LLMs) were used in the preparation of this paper exclusively as general-purpose assistive tools. They supported tasks such as grammar correction, improving readability, and rephrasing for clarity. LLMs were not used to generate original research ideas, design experiments, produce results, or fabricate citations. All technical contributions, experimental designs, analyses, and conclusions in this paper were conceived, implemented, and validated by the authors. The authors remain fully responsible for the accuracy and integrity of the content.

