# OpenReview forum: "DIVERSE: Disagreement-Inducing Vector Evolution for Rashomon Set Exploration"
_ICLR.cc/2026/Conference — ICLR 2026 Poster_

### Official Review · Reviewer_gMcV · 2025-10-24

**Soundness:** 2
**Presentation:** 3
**Contribution:** 2
**Rating:** 4
**Confidence:** 5

**Summary:**

This paper introduce a gradient-free method called  DIVERSE to explore the Rashomon set of deep learning models. This set contains models with similar accuracy but different predictive behaviors. DIVERSE takes a pre-trained model and augments it with "Feature-wise Linear Modulation" (FiLM) layers. DIVERSE then uses a search algorithm "Covariance Matrix Adaptation Evolution Strategy" (CMA-ES) to find different model variations without needing to retrain them from scratch.  The experiments show that DIVERSE can uncover multiple high-performing and functionally distinct models efficiently. It offers a competitive way to explore the Rashomon set, achieving comparable diversity to retraining but at a much lower computational cost.

**Strengths:**

Originality:
The paper reframes latent exploration as Rashomon set exploration in deep networks. It searches a bounded FiLM modulation space around a fixed model and balances an accuracy tolerance epsilon with explicit control over behavioral diversity. This perspective differs from weight generation or full retraining by focusing on efficient, local, and controllable exploration around a single reference model.

Quality:
The methodology is clearly specified and reproducible, with objectives, constraints, and data splits stated in enough detail. The experimental design is sound for the stated goals and uses appropriate datasets and baselines. The evaluation employs complementary metrics including Rashomon Ratio, discrepancy, ambiguity, VPR, and RC, which together provide a comprehensive view of diversity under an accuracy constraint. The analysis includes sensitivity to key hyperparameters and highlights dataset dependent effects.

Clarity:
The paper is clearly written and easy to follow. The flow from problem setup to method and experiments is logical. Notation is consistent, and the figures make the FiLM based search space and the role of the latent variable z intuitive.

Strengths:
The approach is practical and training free for a given reference model, which makes audits of the local Rashomon set feasible under realistic compute budgets. The joint use of decision level and probability level metrics supports a nuanced interpretation of disagreement. By mapping accuracy constrained behavioral variants, the method helps characterize the local performance diversity landscape of a trained model and can inform stress testing, ensembling, and selective prediction. The compute footprint is small compared to retraining, enabling broader exploration on larger models and datasets.

**Weaknesses:**

Scaling of the search. The method relies on full-covariance CMA-ES over FiLM latents, which does not scale well as dimensionality increases. This limits exploration on deeper or more complex models and constrains the approach’s practical reach.

Architecture locality. Because FiLM layers are inserted into a specific pretrained network, the results are tied to that architecture. It is unclear whether conclusions transfer across backbones with comparable accuracy, and cross-architecture comparability is not established.

Objective agnostic to why models disagree. The fitness targets disagreement under an accuracy tolerance but remains insensitive to the underlying cause. As a result, discovered variants may be superficial perturbations rather than models with meaningfully different reliance on features, robustness characteristics, or fairness profiles.

Experimental scope and baseline breadth. Experiments are confined to image classification on moderate-scale datasets. The behavior of the Rashomon set may differ substantially on larger benchmarks (e.g., ImageNet) or in other modalities (e.g., NLP with attention). Moreover, “retraining” is treated as a single comparator despite encompassing diverse regimes, leaving the trade-off landscape underexplored.

**Questions:**

Which layers contribute most to disagreement. Please provide a layerwise or stagewise sensitivity analysis of FiLM norms versus diversity.

How portable is the method across backbones with similar top line accuracy. A compute matched comparison on two architectures would clarify whether results are model local or architecture agnostic.

---

> ### Author Response · Authors · 2025-11-20
>
> We thank the reviewer for the thorough and thoughtful evaluation of our work. We appreciate the constructive comments and respond to each point in detail below.
>
> W1: Your comment encouraged us to investigate alternative CMA-ES variants that might offer better scalability. To the best of our understanding, the FiLM-modulated latent space in DIVERSE is inherently non-separable, since each coordinate of the $z$-vector simultaneously affects many FiLM layers. For such non-separable search landscapes, prior work [1] shows that diagonal or separable variants, such as sep-CMA-ES, tend to perform poorly, as they cannot capture interdependencies between variables.
>
> In exploring further, we also identified dd-CMA-ES [1] as a particularly promising alternative. This variant automatically interpolates between diagonal and full covariance models and has been shown to outperform both on many ill-conditioned or partially non-separable problems. We were unaware of dd-CMA-ES during the development of the method, but agree that it may provide a more scalable optimization strategy for our setting. We will emphasize this in the final version and consider dd-CMA-ES an important direction for future work.
>
> [1] Y. Akimoto, N. Hansen; Diagonal Acceleration for Covariance Matrix Adaptation Evolution Strategies. Evol Comput 2020; 28 (3): 405–435. doi: https://doi.org/10.1162/evco_a_00260
>
> W2: We agree that our current results are tied to the specific pretrained architectures into which FiLM layers were inserted. Since DIVERSE explores the local Rashomon set around a fixed reference model, the discovered variants naturally reflect architectural properties of that model, and cross-architecture comparability is not directly established in our experiments. Our claims are therefore grounded strictly in the architectures we evaluate, and we are careful not to overclaim generality beyond these settings.
>
> This architectural locality follows directly from the design goal of DIVERSE: to provide an efficient, training-free method for local exploration around a single pretrained model. Each run of DIVERSE characterizes the Rashomon landscape of that model rather than across different backbones. We view this as a first step rather than a restriction of the method itself. In principle, DIVERSE can be applied to any pretrained model for which FiLM insertion points can be defined, including multiple backbones with comparable accuracy.
>
> W3: Our objective in this work is to systematically recover and characterize the local Rashomon set, i.e., the collection of models that differ in their predictive behavior while remaining within a strict accuracy tolerance. The fitness function is therefore designed to surface diverse, $\epsilon$-optimal variants, without committing to a specific mechanism behind their disagreement.
>
> At the same time, we agree that understanding how and why these models disagree is highly valuable. In line with Reviewer CZF8’s helpful suggestion, we will add qualitative illustrative examples showing how multiple $\epsilon$-optimal models classify the same input differently. These examples make behavioral differences concrete and help readers intuitively assess whether disagreements appear superficial or reflect deeper changes in model behavior. While this is not a full causal analysis, it provides an important first step toward interpreting the variability that DIVERSE uncovers.
>
> A deeper causal analysis of why models disagree, e.g., in terms of feature reliance or robustness, is an important direction for future work beyond the scope of this methodological contribution.
>
> W4: Our experimental scope is intentionally limited to image classification on moderate-scale datasets, as our focus in this paper is to introduce DIVERSE and validate the feasibility of FiLM-based Rashomon set exploration in deep neural networks. Extending DIVERSE to larger benchmarks such as ImageNet or to other modalities (e.g., NLP) should be possible, but doing so requires additional architectural considerations, particularly in non-convolutional models such as Transformers, where it is not yet clear which components are most appropriate for FiLM-style modulation. Investigating these design choices is therefore an important direction for future work, but lies beyond the scope of this initial methodological contribution.
>
> Regarding baselines, we evaluated DIVERSE against both retraining and the recent dropout-based Rashomon exploration method. Retraining serves as the standard, widely used comparator for Rashomon analysis, while the dropout baseline provides a lightweight, training-free alternative. Our aim was not to exhaustively enumerate all retraining regimes, but to compare against the two approaches most relevant to the Rashomon literature. Broadening this comparison to additional datasets, modalities, and retraining variants could be a valuable extension.

---

> ### Author Response · Authors · 2025-11-20
>
> Q1: To identify which FiLM layers contribute most to disagreement, we performed a layerwise ΔTVD sensitivity analysis on the Rashomon set. After CMA-ES discovers a set of $\epsilon$-Rashomon members with all FiLM layers active, we take each member’s latent vector $z$ and measure its baseline disagreement with the reference model using Total Variation Distance. We then disable one FiLM layer at a time, recompute the model’s predictions, and record how much the disagreement changes when that layer is removed.  A positive ΔTVD indicates that the FiLM layer generates diversity (removing it reduces disagreement), while negative ΔTVD indicates the layer suppresses diversity. All FiLM layers in a given model are driven by the same latent vector and use identical initialization, so differences in ΔTVD reflect only the architectural location of each FiLM site. Averaging ΔTVD across all Rashomon members yields a clear functional importance ranking.
>
> Across all three architectures, we find that disagreement is localized: early FiLM layers dominate in MNIST, mid-level convolutional FiLMs dominate in VGG16, and early/mid-stage FiLMs dominate in ResNet50, while several deeper FiLM sites consistently suppress disagreement.
>
> An interesting direction for future work could be to leverage the ΔTVD sensitivity scores to automatically select FiLM sites, allowing CMA-ES to explore only the layers that contribute meaningfully to functional diversity. We will incorporate the full sensitivity analysis in the final version of the paper, with the corresponding figures included in the main text or the appendix depending on space constraints.
>
> Q2: Our current results are architecture-local because each run of DIVERSE explores the Rashomon set around a single pretrained backbone. This is intentional: we aimed to establish the method on representative CNN/MLP architectures without overclaiming cross-architecture generality.
>
> In principle, DIVERSE can be applied to other backbones with similar accuracy, but a compute-matched comparison would require additional architectural decisions (e.g., FiLM placement per backbone) and is affected by the CMA-ES scaling limitations noted earlier. As a result, evaluating portability across architectures is an interesting and important direction for future work, but lies outside the scope of this initial methodological contribution.

---

> > ### Comment · Reviewer_gMcV · 2025-11-22
> >
> > Thank you for the rebuttal and the clarifications. Overall, I see W1, W2, W3, and W4 as only partially addressed, Q1 as fully addressed, and Q2 as partially addressed. I still have two main concerns. First, since the rebuttal now points to dd-CMA-ES as a promising way to improve scalability, it is not clear why it cannot be implemented and tested on at least one of the existing models (and possibly a slightly larger one) during the discussion period to provide some concrete evidence on scaling.
> >
> > Second, while the method is described as, in principle, applicable to any architecture, applying it to non-convolutional models such as Transformers is said to require additional structural changes. This seems unclear to me or a little confusing.
> > Given that modern Transformers are modular stacks of self-attention and MLP blocks, and FiLM is already applied to simple MLPs in your experiments, it seems feasible to at least probe FiLM insertion in a small Transformer, so I would encourage either softening the generality claim or more clearly validate your claim  empirically.

---

> > > ### Author Response · Authors · 2025-12-03
> > >
> > > We want to thank the reviewer for their time and thoughtful feedback.
> > >
> > > Q1: We fully agree that DD-CMA-ES is a promising direction for improving scalability. However, given the short discussion period and the implementation effort required to properly integrate, tune, and validate DD-CMA-ES , we felt that adding a partially verified variant would risk introducing instability rather than strengthening the paper. Because of this, we faced a choice between (i) attempting to optimize our method using DD-CMA-ES, or (ii) conducting a new experiment on a different architecture class. Since one of our main goals is to assess the generality of DIVERSE, we prioritized evaluating whether our approach also extends to more complex architectures such as Vision Transformers. We therefore opted to run a focused ViT experiment, which we believe provides more directly relevant evidence for the method’s breadth. We will highlight DD-CMA-ES as an important direction for future work.
> > >
> > > Q2: We appreciate the reviewer’s suggestion to apply FiLM only to the MLP blocks of a Transformer, which aligns well with our MLP insertion strategy. Motivated by this, we conducted a new experiment on a compact Vision Transformer trained on CIFAR-10, using FiLM exclusively on the dense layers inside the Transformer MLP blocks and the classifier head (Appendix A.6). The resulting Rashomon ratios and diversity metrics are comparable to those of VGG-16 at the same latent dimensionality, and runtime is similar as well. These findings, although preliminary and limited to $d = 2$, provide encouraging evidence that DIVERSE can extend to Transformer architectures. A more comprehensive study, including a full ablation over FiLM insertion sites and larger latent dimensions, is needed to fully verify the integration of the FiLM modulation into the Transformer architecture.

---

### Official Review · Reviewer_HCic · 2025-10-31

**Soundness:** 3
**Presentation:** 4
**Contribution:** 2
**Rating:** 6
**Confidence:** 3

**Summary:**

The authors introduce DIVERSE, a novel method to find other models in the ***Rashomon Set*** of a reference model. A *Rashomon Set* is defined as the set of models that have similar predictive accuracy on a given task.

The DIVERSE algorithm is based on two main components: FiLM layers which apply affine transformations to pre-activations, and CMA-ES for gradient-free optimisation of the parameters of the FiLM layers. Since CMA-ES is not scalable to high dimensions, DIVERSE uses it to optimise a low-dimensional vector \$z\$, which is then projected to higher dimensions using random fixed matrices.

The authors evaluate their method using both prediction-based and probability-based metrics, in all cases after the last layer. They compare against two baselines: re-training using different seeds, and dropout-based Rashomon exploration. In terms of diversity, the generated models are generally less diverse than with full re-training, but for a runtime orders of magnitude lower.

**Strengths:**

This paper proposes a relatively simple, inexpensive and elegant solution to the Rashomon Set exploration problem.

S1: The proposed method is relatively simple, yet effective. This makes it very relevant to solve the Rashomon Set exploration problem.

S2: The evaluation of the algorithm is good, with metrics covering both class predictions and output probabilities. Furthermore, multiple hyperparameters are explored and DIVERSE is compared to existing baselines.

S3: The paper is generally clear and well-written. The introduction effectively contextualises the Rashomon Set problem, experimental setup and results are generally clear.

**Weaknesses:**

I think that the paper is overall quite solid. However, I believe its main weak points are related to its impact and motivation.

W1: Based on the Introduction and the Conclusion of the paper, I do not understand why Rashomon Set exploration is an important problem to solve. I would like the authors to better motivate why their work is impactful for research in Machine Learning.

W2: Furthermore, I think the results insufficiently show that DIVERSE is better than dropout-based Rashomon exploration. In particular, dropout-based Rashomon exploration is less computationally expensive than DIVERSE, and shows better performance in most cases for PneumoniaMNIST. I believe the authors should better highlight where their method outperforms the existing baselines, and should clarify under what conditions DIVERSE is preferable, such as specific datasets, architectures, or computational constraints.

W3: For the experiments comparing DIVERSE with its baselines, it is unclear to me what hyperparameters are being used. I think this should be described in the experiment setup.

W4: This is a minor point, but I believe it would be preferable that the metric mathematical definitions, currently in Appendix A.1, be integrated to the main text. This would improve the clarity and readability of the results Section.

**Questions:**

Q1: The definition in the Introduction defines the Rashomon Set as the set of models that achieve a similar performance on a same task, whereas in Equation (1) this set is constrained to a hypothesis space of models parametrised by weights \$w \in \mathbb{R}^p\$. Could the authors please clarify whether the Rashomon Set is constrained to models that use the same architecture?

Q2: In Table 1, it is unclear which size \$d\$ is used for the vector \$z\$. Since CMA-ES struggles with higher dimensionalities, how do higher values of \$d\$ impact the runtime of DIVERSE? Is there a trade-off between diversity, Rashomon ratio and runtime as \$d\$ increases?

Q3: The paper only explores using DIVERSE on CNNs. Would DIVERSE be applicable to Transformers? If yes, demonstrating this applicability in the paper could further strengthen it. If not, what are the potential challenges?

---

> ### Author Response · Authors · 2025-11-20
>
> We thank the reviewer for the careful reading of our paper and for highlighting its strengths. We appreciate the constructive feedback and address each remark below.
>
> W1: Thank you for highlighting the need to better motivate the importance of Rashomon set exploration. We fully agree, and we will revise the Introduction and Conclusion to make this central motivation clearer.
>
> Predictive multiplicity, a manifestation of the Rashomon Effect, implies that two models with nearly identical overall performance may still disagree on specific individual samples. In practice, this means that end users can receive contradictory predictions from models that should be equally reliable. In high-risk scenarios such as PneumoniaMNIST, such contradictions are especially concerning. Importantly, these disagreements are not noise or model error, but legitimate alternative decision boundaries that all lie within the $\epsilon$-optimal region of the loss landscape.
>
> Therefore, methods for systematically generating Rashomon members are essential: they allow us to reveal the space of plausible yet contradictory predictions, and understand where models diverge. While this has been explored for simpler model families [1, 2], Rashomon exploration is extremely challenging for deep neural networks due to their high-dimensional hypothesis spaces and the computational cost of retraining-based methods.
>
> DIVERSE provides a computationally feasible approach to explore this space in modern DNNs. Following Reviewer CZF8’s helpful suggestion, we will include illustrative examples showing how multiple $\epsilon$-optimal models, generated by DIVERSE, classify the same input differently. These examples concretely demonstrate predictive multiplicity and highlight why it matters for end users. We will further discuss these visual results in the Discussion section, clarifying what such divergent yet equally accurate predictions can imply for downstream decision-making and practical end-user systems
>
> [1] Rui Xin, Chudi Zhong, Zhi Chen, Takuya Takagi, Margo Seltzer, and Cynthia Rudin. Exploring the whole rashomon set of sparse decision trees. In Proceedings of the 36th International Conference on Neural Information Processing Systems, NIPS ’22, Red Hook, NY, USA, 2022. Curran Associates Inc. ISBN 9781713871088.
>
> [2] Chudi Zhong, Zhi Chen, Jiachang Liu, Margo Seltzer, and Cynthia Rudin. Exploring and interacting with the set of good sparse generalized additive models. In Proceedings of the 37th International Conference on Neural Information Processing Systems, NIPS ’23, Red Hook, NY, USA, 2023.Curran Associates Inc.
>
>
> W2: We agree that dropout-based Rashomon exploration is extremely fast, and on PneumoniaMNIST it achieves higher diversity under the authors’ evaluation protocol. However, a key methodological distinction is that the dropout approach applies stochastic dropout masks directly on the test set, and defines Rashomon membership on that same test set. This procedure makes their diversity estimates optimistic, since model selection and evaluation occur on the same data.
>
> This observation in fact motivated us to adopt a more robust and principled evaluation pipeline: (i) CMA-ES explores the latent space only on the training set, (ii) Rashomon membership is checked on a held-out validation set, and (iii) all multiplicity metrics are reported solely on the test set. This three-stage procedure minimizes selection bias and ensures that reported Rashomon members genuinely generalize beyond the data used for search. We believe this rigor is especially important in sensitive or high-risk settings such as PneumoniaMNIST.
>
> Because DIVERSE evaluates candidates on both the training and validation sets, which are often larger than the test set, the total time to generate $m$ models is naturally longer than dropout. The additional cost reflects the principled separation between search, selection, and evaluation. Under this protocol, DIVERSE often outperforms dropout, especially on CIFAR-10 and MNIST, and remains competitive on PneumoniaMNIST at higher $\epsilon$-levels.

---

> ### Author Response · Authors · 2025-11-20
>
> W3: Thank you for pointing this out, we agree that the hyperparameter choices for the baselines should be stated more explicitly. In our experiments, we evaluated multiple hyperparameter settings for each dataset and each Rashomon threshold $\epsilon$. For DIVERSE, we then selected the best-performing configuration per dataset, where “best-performing” refers to the hyperparameter set that showed the most consistent and stable performance across all reported metrics.
>
> This provides a fair and representative comparison with the baselines, while avoiding sensitivity to individual runs or specific hyperparameter choices. In the final version of the paper, we will update the Baselines paragraph in Section 4 to explicitly list the exact hyperparameters used for each dataset, including CMA-ES step size, latent dimensionality, $z_0$ value, and the epsilon used.
>
> W4: We agree that including the mathematical definitions of the metrics in the main text would improve readability. However, we are constrained by the page limit, and moving the full set of definitions from Appendix A.1 into the main body would require removing other essential methodological details. To remain transparent and complete, we chose to provide the full formal definitions in the appendix while keeping concise descriptions in the main text.
>
> Q1: We appreciate the opportunity to clarify the definition. In our work, the Rashomon set definition is not conceptually restricted to a single architecture. In principle, the Rashomon set may even be heterogeneous, containing models of different architectures, training procedures, or inductive biases, as long as they achieve performance within the $\epsilon$-tolerance of the reference model.
>
> Equation (1) follows the standard supervised learning formulation, where models are written as $f_w \in \mathcal{H}$ for notational convenience. In our experimental setting, we focus on exploring the local Rashomon set around a specific pretrained architecture, because DIVERSE is designed to investigate functional variations induced through FiLM modulation of a fixed backbone. However, the general definition is not tied to a particular architecture, and we will revise the first paragraph of the Introduction to make this distinction clearer.
>
> Q2: We will clarify in Table 1 which latent dimensions $d$ correspond to the reported runtimes. Because we use CMA-ES, the number of evaluated models is not an arbitrary choice but is determined by the default CMA-ES settings, specifically the population size and the number of generations. Both of these scale with the dimensionality of the latent vector $z$. Higher-dimensional search requires larger populations and more generations to adequately explore the space, which naturally increases the total runtime. This follows the standard behavior of full-covariance CMA-ES, whose update operations grow more expensive as $d$ increases.
>
> The effects of $d$ on performance are visible in Figure 2. On MNIST, Rashomon members are found across all combinations of $\epsilon$ and $d$, although diversity gains diminish at larger $d$, and the Rashomon Ratio becomes unstable under very strict $\epsilon$. In contrast, PneumoniaMNIST and CIFAR-10 only admit Rashomon sets at small latent dimensions; for $d \geq 16$, CMA-ES fails to find candidates that satisfy the Rashomon constraint. This reflects the more complex search landscapes of deeper networks with many FiLM injection points: larger latent spaces make it more difficult for CMA-ES to remain within the narrow $\epsilon$-tolerance.
>
> Overall, the results show a trade-off: increasing $d$ expands the expressive capacity of possible FiLM modulations but makes the optimization problem harder.
>
> Q3: In principle, DIVERSE is not restricted to convolutional architectures. The method requires a couple of components, such as a pretrained reference model and locations in the network where FiLM-like affine modulations can be inserted. Both of these conditions should be equally achievable for Transformer architectures.
>
> That said, applying DIVERSE to Transformers introduces several design challenges that require careful consideration. The main difficulty, in our view, is deciding where FiLM layers should be placed. In CNNs, FiLM operates on channel-wise feature maps. In Transformers, the architecture is structurally different, and one must determine which part of the network should be modulated by FiLM. Exploring different insertion strategies requires an additional architectural study to identify stable and meaningful FiLM insertion points. We view this as promising future work and an important step forward.

---

> > ### Comment · Reviewer_HCic · 2025-11-25
> > **Rebuttal follow-up**
> >
> > I thank the authors for their complete and detailed response. As a preamble, do the authors plan to post a revision of their submission during the discussion period? Reviewing it would be valuable in assessing how the authors addressed the various weaknesses highlighted by the reviewers. The authors mentioned several changes that I believe will enhance the quality of the paper, and I am eager to see these reflected in the revised version.
> >
> > In the meantime, I already have a few follow-up questions regarding the rebuttal by the authors.
> >
> > W1: Many thanks for this explanation, it makes the motivation of Rashomon Set exploration clearer to me. I look forward to reading the expanded Introduction and Discussion sections of the paper.
> >
> > W2: Thank you for clarifying methodological differences between dropout-based Rashomon exploration (hereafter DBRE, for brevity) and DIVERSE. The fact that DBRE runs specifically on test data is interesting. I have a few follow-up questions that arise from this:
> >
> > (i) This difference in setting makes it difficult to compare DIVERSE with DBRE in a fair way. Currently, the authors have chosen to directly apply DBRE on the test data, while DIVERSE is an algorithm which bases itself on the training data. In order to make the comparison fairer, do you think it would make sense to evaluate DBRE in another experimental setup where model selection happens on training data, and evaluation on test data? This would help understanding how well DBRE dropout masks generalise to unseen data. Furthermore, I hypothesise that results will be less favourable to DBRE, which would strengthen the claim that DIVERSE is a better solution.
> >
> > (ii) In the original DBRE paper, the authors suggest using their algorithm together with re-training to make the search space of the Rashomon Set more diverse, since DBRE only computes dropout masks based on a given trained model. Both DIVERSE and DBRE are cheap to compute: perhaps they could be combined? In that case, would a method combining both solutions outperform DBRE, DIVERSE, or both?
> >
> > W3: Thank you for your response. I look forward to this change in the revision, it will make comparing results between the different Figures and Tables easier.
> >
> > W4: I understand that the authors can be limited in that regard by the page limit. As per the author instructions, the page limit is increased from 9 to 10 pages during the discussion/revision period. Depending on how much content the authors have to add to the revision based on the reviewers’ feedback, I believe that adding the mathematical formulations to the main text would be valuable. If there is not enough space because of the content added during discussion, I agree that this is secondary and that definitions can stay in the Appendix.
> >
> > Q1: Thank you, this makes the definition of the Rashomon Set clearer to me. I believe that based on the authors’ response, it would make sense to clarify the main claim of the paper, for example on line 53, and to make explicit that the authors explore a *local* Rashomon Set, based on a pre-trained model’s architecture and weights.
> >
> > Q2: Thank you for this clarification. I agree with the authors that Figure 2 illustrates the trade-off between expressiveness and performance of using a higher dimensional latent vector. I have read through the methods again, and see that the authors define $m = k \times d$ with $k = 80$. This makes the link between $m$ and $d$ explicit, and I agree with the authors’ suggestion of adding the value of $d$ to Table 1, as it would help readers build intuition. I am still confused about the meaning of the variable $k$, but I think this comes from my own lack of familiarity with CMA-ES. From my understanding, for every generation, $\textrm{popsize} = 4 + 3 \log d$ candidates are being generated, and it naturally follows that the number of generations is set to $m / \textrm{popsize}$. I have difficulties understanding, however, why $m = k \times d$, why $k = 80$, and what $k$ represents exactly.
> >
> > Q3: Regarding this question, I agree with reviewer gMcV that DIVERSE should theoretically be applicable to the MLP layers within the Transformer architecture without much change. I think that showing whether DIVERSE works in that setup would strengthen this paper by demonstrating validity beyond MLP and CNN architectures.

---

> > > ### Author Response · Authors · 2025-12-03
> > >
> > > We thank the reviewer for their thoughtful and constructive feedback. We will upload a revised version of our submission that incorporates all of the points raised during the discussion period.
> > >
> > >
> > > W2: We agree that applying DBRE under a training/validation/test protocol, analogous to the one used by DIVERSE, would provide a fairer comparison and would likely yield less optimistic diversity estimates for DBRE, since model selection would no longer occur on the test set. This is an insightful suggestion. However, due to time constraints during the rebuttal period, we were unable to run a full set of DBRE-on-validation experiments. For completeness and comparability with prior work, we therefore report DBRE exactly as defined in its original paper.
> > >
> > > We appreciate the reviewer’s suggestion of combining DBRE with DIVERSE. This is an interesting direction; however, it targets a broader design space than we aimed to address in this work, which focuses specifically on FiLM-based latent modulation. A thorough investigation of such a hybrid method would require additional methodological development and experimentation, and we consider it beyond the scope of the present study.
> > >
> > >
> > > Q2: In our CMA-ES setup, $k$ specifies the number of evaluations per latent dimension, giving a total budget $m = k d$. While CMA-ES has no strict rule for this choice, community practice often uses budgets of tens to hundreds of evaluations per dimension; our setting of $k=80$ aligns with this and keeps computation tractable. With a population size of $\text{popsize} = 4 + 3 \log d$, this budget corresponds to $n_{\text{generations}} = \lceil k d / \text{popsize} \rceil$ generations, after which the search stops. The resulting number of evaluated models is $\text{popsize} \cdot n_{\text{generations}}$, differing from $k d$ by at most one additional generation. We further motivate and elaborate on this in Section 4 of the revised submission.
> > >
> > >
> > > Q3:
> > > Motivated by reviewer gMcV, we conducted a new experiment on a compact Vision Transformer trained on CIFAR-10, using FiLM exclusively on the dense layers inside the Transformer MLP blocks and the classifier head (Appendix A.6). The resulting Rashomon ratios and diversity metrics are comparable to those of VGG-16 at the same latent dimensionality, and runtime is similar as well. These findings, although preliminary and limited to $d = 2$, provide encouraging evidence that DIVERSE can extend to Transformer architectures. A more comprehensive study, including a full ablation over FiLM insertion sites and larger latent dimensions, is needed to fully verify the integration of the FiLM modulation into the Transformer architecture.

---

### Official Review · Reviewer_CZF8 · 2025-10-31

**Soundness:** 2
**Presentation:** 3
**Contribution:** 2
**Rating:** 6
**Confidence:** 2

**Summary:**

The paper adresses the problem of exploring the Rashomon set of a trained machine learning model in a cost-efficient and diverse manner. The $\epsilon$-Rashomon set is the set of all machine learning models that reach the same empirical risk as the reference model with a tolerance $\epsilon$. In this set lies a multiplicity of different models that may produce different predictions for identical individuals. Being able to explore $\epsilon$-Rashomon sets model is interesting to construct diverse ensemble of machine learning models for uncertainty quantification or improved predictive performance.

As exploring the $\epsilon$-Rashomon set of a deep neural network can be compute intensive with naive methods (retraining from scratch or exploring the whole parameter space), this paper proposes to explore a subspace of the $\epsilon$-Rashomon set using Feature-wise Linear Modulation (FILM), a low-dimensional parameterized transformation of a neural network. In this low dimensional space, a black-box and derivative free optimization algorithm is used (CMA-ES) for exploration. The constraint on exploring models included in the $\epsilon$-Rashomon set is relaxed by adding a penalization in the objective function minimized by CMA-ES.

The proposed approach (DIVERSE) is then evaluated on three small to medium scale image classification datasets. Empirical results show that DIVERSE can be a good compromise between exploration and compute. Extensive experiments and ablation studies are conducted to evaluate the impact of each introduced hyperparameters.

**Strengths:**

1. The paper is well-written with a clear objective, all notions are introduced clearly.
2. The proposed approach has the potential to be a fundamental tool in many domains of machine learning.
3. The proposed approach is sound and well grounded in the literature.
4. The ablation study is quite furnished and the experimental protocol sound

**Weaknesses:**

1. The paper lacks qualitative or illustrative experiments to helps the reader gain intuitions on the significance of the results.
2. The paper lacks experiments with downstream tasks (uncertainty quantification, ensembling, ...) to better asses if generated models with DIVERSE are actually effective for practical tasks.
3. The paper lacks quantification about how smaller the explored set of models induced by FiLM is to the true $\epsilon$-Rashomon set.

**Questions:**

Has Rashomon set exploration been done for different training tasks such as regression?

### Remarks
The columns of Figure 2 and 3 are not in the same order.
The impact of $\lambda$, the mixing coefficient between soft and hard agreement is not evaluated in the ablation study.
More diverse datasets could be used in the experimental protocol. MNIST (and I think PneumoniaMNIST also) is quite an easy dataset where a linear model trained on the raw pixel can achieve very high accuracy. Even though not ideal, Fashion-MNIST and K-MNIST might be better alternatives. MNIST-1D, even though not a image classification dataset but a time-series classification one, could be a potential candidate, as linear model have poor performances on it.
- Fashion-MNIST : Xiao, Han, Kashif Rasul, and Roland Vollgraf. "Fashion-mnist: a novel image dataset for benchmarking machine learning algorithms." 2017.
- K-MNIST : Clanuwat, Tarin, et al. "Deep learning for classical japanese literature." 2018.
- MNIST-1D : Greydanus, Sam, and Dmitry Kobak. "Scaling down deep learning with mnist-1d." 2020.

---

> ### Author Response · Authors · 2025-11-20
>
> We thank the reviewer for the thorough evaluation, constructive remarks, and valuable suggestions. Below we address each point in detail and clarify the aspects raised.
>
> W1: We thank the reviewer for the suggestion. To provide qualitative intuition, we will add visual examples that show how different Rashomon members interpret the same input. For each dataset (MNIST, CIFAR10, PneumoniaMNIST), we highlight the samples with highest model disagreement and display (i) the image itself and (ii) a class-frequency distribution over all FiLM-modulated models. This "per-sample disagreement visualisation" shows how many models predict each class, revealing groups of distinct predictions and making functional diversity immediately apparent. These qualitative panels demonstrate how different models in the Rashomon set focus on different plausible explanations for the same data point. We will include 1–2 such examples per dataset (within page limits) in the final version.
>
> W2: Our goal in this work is to systematically explore Rashomon sets in deep neural networks rather than to optimize ensembles directly. However DIVERSE should naturally be compatible with downstream tasks such as ensembling and uncertainty quantification. Recent work on Rashomon-based ensembles [1] shows that selecting diverse, near-optimal models can improve robustness and calibration; although their approach is tree-based, the underlying idea should transfer directly.
>
> To adapt DIVERSE for ensembles would require modifying the fitness function, for example, by encouraging (i) the coverage of different predictive modes and (ii) low cross-model correlation, while still maintaining the $\epsilon$-loss constraint. CMA-ES would then search for a set of FiLM-modulated models optimized for not only diversity but also for ensemble performance and no other architectural changes would be required. We will clarify this in the final version and see downstream ensemble applications as an exciting direction for future work. How such Rashomon-derived model sets are ultimately incorporated into real-world systems is an active area of research, but ensembles are a particularly promising avenue.
>
> [1] Gianlucca Zuin and Adriano Veloso. 2025. “A 6 or a 9?”: Ensemble Learning through the Multiplicity of Performant Models and Explanations. ACM Trans. Knowl. Discov. Data 19, 9, Article 172 (November 2025), 39 pages. https://doi.org/10.1145/3767735
>
> W3: Unfortunately, the size of the true $\epsilon$-Rashomon set of a deep neural network is not measurable in practice. As shown in [2], Rashomon sets are often extremely large even for simple linear models, and due to the structure of neural networks that set should be too large to enumerate or quantify. To the best of our knowledge no related work has computed the size of the full Rashomon set for deep networks, for that reason we follow the recent dropout-based method [3]. And operate on the empirical Rashomon sets, which are strict subsets and provide lower bounds on predictive multiplicity compared to the true $\epsilon$-Rashomon set.
>
> [2] Semenova, Lesia & Rudin, Cynthia & Parr, Ronald. (2022). On the Existence of Simpler Machine Learning Models. 1827-1858. 10.1145/3531146.3533232.
>
> [3] Hsiang Hsu, Guihong Li, Shaohan Hu, and Chun-Fu Chen. Dropout-based rashomon set exploration for efficient predictive multiplicity estimation. In The Twelfth International Conference on Learning Representations, 2024.
>
> Q1: Yes, Rashomon set exploration has been studied in regression settings. Prior work [4, 5, 6] has analyzed Rashomon sets for linear and tree-based regression models, showing that large sets of functionally similar predictors can coexist under standard regression losses. While our experiments focus on classification, DIVERSE is not restricted to this setting: applying it to regression would require replacing the cross-entropy penalty with a regression loss (e.g., MSE) and using regression-appropriate disagreement metrics (e.g., mean absolute deviation or variance across predictions). The FiLM-modulated latent search and CMA-ES optimization remain unchanged. We will clarify this in the final version, and view extending and fully validating DIVERSE in regression settings as an important direction for future work, which will require additional targeted experiments.
>
> [4] Fisher, Aaron & Rudin, Cynthia & Dominici, Francesca. (2019). All Models are Wrong, but Many are Useful: Learning a Variable's Importance by Studying an Entire Class of Prediction Models Simultaneously. Journal of machine learning research : JMLR. 20.
>
> [5] Semenova, Lesia & Rudin, Cynthia & Parr, Ronald. (2022). On the Existence of Simpler Machine Learning Models. 1827-1858. 10.1145/3531146.3533232.
>
> [6] Semenova, Lesia & Chen, Harry & Parr, Ronald & Rudin, Cynthia. (2023). A Path to Simpler Models Starts With Noise. Advances in neural information processing systems. 36. 3362-3401.

---

> ### Author Response · Authors · 2025-11-20
>
> R1: Thank you for noting the figures having different column orders, this will be fixed in the final version.
>
> R2: Thank you for this observation. Following the reviewer’s comment, we performed a targeted $\lambda$-ablation over $\lambda \in$ {0.0, 0.25, 0.5, 0.75, 1.0} on all three datasets. We observed extremely small to no variation in Rashomon Ratio, Ambiguity, Discrepancy, VPR, or Rashomon Capacity. Although changing $\lambda$ does change the numerical diversity score and therefore the numerical fitness values, in our FiLM-modulated models soft (TVD-based) and hard (label-based) disagreement are highly correlated. As a result, varying $\lambda$ does not meaningfully change the ranking of candidates within each CMA-ES generation. Since CMA-ES is rank-based, identical rankings lead to identical search trajectories and thus to the same discovered Rashomon members.
>
> We use $\lambda$ because it makes explicit how the two components of our diversity measure, probability-level shifts (TVD) and label-level changes, are combined. This provides a transparent and interpretable formulation of the objective and allows us to assign a balanced weighting; for this reason, we adopt $\lambda$ = 0.5 in the main experiments. We will clarify this in the revision and include the full $\lambda$-sweep results in the appendix.

---

### Official Review · Reviewer_mafH · 2025-11-01

**Soundness:** 3
**Presentation:** 3
**Contribution:** 4
**Rating:** 8
**Confidence:** 3

**Summary:**

The authors introduce DIVERSE, a framework for exploring the Rashomon set of deep neural networks, innovative way to find high quality and diverse models that match a reference model's accuracy but differ in their predictions. DIVERSE adds Feature-wise Linear Modulation (FiLM) layers to a pretrained model and uses Covariance Matrix Adaptation Evolution Strategy (CMA-ES) to search a latent modulation space, producing diverse model variants without retraining or gradients. On MNIST, PneumoniaMNIST, and CIFAR-10, DIVERSE finds multiple high-performing models that behave differently

**Strengths:**

- Innovative approximation of Quality-Diversity evolution
- Well written paper

**Weaknesses:**

- Limited Ablation study on evolution side.

**Questions:**

Why CMA-ES and not any other evolutionary algorithm?

---

> ### Author Response · Authors · 2025-11-20
>
> We thank the reviewer for the positive and thoughtful assessment of our work. We appreciate the encouraging remarks and are happy to address the points raised.
>
> W1: We agree that it is important to verify that our results are not an artifact of a particular evolutionary setup. In practice, CMA-ES has very few effective hyperparameters; the main problem-dependent choices are the latent vector initializations and the initial step-size $\sigma_0$ [1]. ​In our experiments, we ablated both: we ran DIVERSE from multiple initial latent vectors and varied $\sigma_0$  in the range [0.1, 0.5] which spans conservative to relatively aggressive initial exploration. The results of these ablations are reported in Appendix Figures 4 and 5, which consistently show stable diversity–accuracy behavior across all tested settings. We acknowledge that a broader sweep over $\sigma_0$ (e.g., including more extreme values) could be added for completeness, and we view this as a natural extension, but the current ablation already suggests that DIVERSE does not rely on a finely tuned CMA-ES configuration.
>
> Q1: We chose CMA-ES, after careful consideration, because our latent space is small ($d$ ≤ 64) and strongly correlated: each dimension of the vector z influences many FiLM layers simultaneously, which creates a non-separable search landscape. Prior work shows that full covariance CMA-ES is the safest and most reliable choice in such settings [2, 3]. In particular, CMA-ES maintains rotational invariance and can adapt its search distribution to follow correlated directions, a key property when dimensions interact as they do under FiLM modulation. Moreover, a comparative study has shown that CMA-ES consistently outperforms alternative evolutionary algorithms such as Differential Evolution and Particle Swarm Optimization on low-to-moderate dimensional and non-separable problems [2], which matches our scenario. We will clarify these points in Section 2 of the final version of the paper.
>
> [1] Hansen, Nikolaus. (2007). The CMA Evolution Strategy: A Comparing Review. 10.1007/3-540-32494-1_4.
>
> [2] Auger, A., Hansen, N., Perez Zerpa, J. M., Ros, R., & Schoenauer, M. (2009). Experimental comparisons of derivative free optimization algorithms (invited talk). In Experimental Algorithms - 8th International Symposium, SEA 2009, Proceedings (pp. 3-15). (Lecture Notes in Computer Science (including subseries Lecture Notes in Artificial Intelligence and Lecture Notes in Bioinformatics); Vol. 5526 LNCS). https://doi.org/10.1007/978-3-642-02011-7_3
>
> [3]  Omidvar, Mohammmad Nabi & Li, Xiaodong. (2010). A Comparative Study of CMA-ES on Large Scale Global Optimisation. Proc. Advances Artif. Intell.. 6464. 303-312. 10.1007/978-3-642-17432-2_31.

---

### Author Response · Authors · 2025-12-03
**Revised Version Uploaded**

We have uploaded a revised version of the paper reflecting the clarifications and revisions discussed in our responses.

---

### Meta-Review · Area_Chair_h456 · 2026-01-12

**Summary:**

Overall, this paper provides a neat solution to a well-defined problem. Reviewers have discussed weaknesses, e.g. the lack of downstream tasks, comparisons to dropout-based methods, scaling and specificity to a particular trained neural network. Most of these have been addressed to a degree, but it does seem that the discussions have been cut short. Several changes were made to the paper, although the reviewers did not directly comment on this. Overall, this seems to be a paper that the reviewers appreciated, but with the discussion cut short, it is borderline.

**Reviewer Concerns:**

The point on evaluation, particularly when compared to dropout-based methods, were answered in a clear and convincing way.
Scalability issues were discussed, and possible solutions were proposed, but actual implementation and testing were postponed to later.
The issue of the results not being used in downstream tasks was explicitly rejected as not applicable by the authors, and left for the future. This remains a weakness in my opinion.

**Reviewer Scores:**

It is possible that gMcV may have increased their score, had the discussion phase lasted longer. They engaged in the discussion well, and acknowledged that certain issues had been addressed.

---

### Decision · Program_Chairs · 2026-01-26

Accept (Poster)